# A Survey on Hallucination in Video Understanding: Taxonomy, Causes, and Mitigation Techniques

## Abstract

Video Large Language Models (Vid-LLMs) have recently achieved strong performance across a wide range of video understanding tasks, including question answering, captioning, and multimodal reasoning. However, these models frequently produce outputs that are not faithfully grounded in the underlying video content, a phenomenon commonly referred to as hallucination. Compared with hallucination in text-only or image-based models, hallucination in video understanding is further complicated by temporal dynamics, motion interpretation, long-context dependencies, and event-level reasoning. In this survey, we present a comprehensive review of hallucination in Vid-LLMs. We begin with a unified taxonomy, which categorizes different hallucination phenomena, summarize dataset for each hallucination type, and then organize existing mitigation strategies according to the failure mechanisms they address. We also discuss key open challenges and outline some promising research directions.

## 1 Introduction

AI models are increasingly deployed in real-world applications requiring perception, reasoning, and decision-making at scale. Video understanding has emerged as a core task for modern multimodal foundation models. These systems take dynamic visual streams as input, recognize objects and actions, track interactions, and reason about events over time. Video understanding systems are now widely adopted in high-stakes settings, including but not limited to: surveillance and detection, autonomous driving, human-robot interactions, and large-scale video content moderation (Chen et al., 2024b; Eze & Crick, 2025; Wang et al., 2025p).

Incorrect or unsupported interpretations of video content, which are commonly referred to as **hallucinations**, can result in degraded system performance and serious safety, legal, or ethical consequences (Wang et al., 2025l). Nonexistent actions, misidentified objects, or fabricated event explanations may propagate to downstream decision-makers, amplifying their impact beyond isolated errors (Wang et al., 2025n). Rather than viewing hallucination in video understanding merely as a modeling imperfection, we should recognize it as a system-level reliability issue that directly affects trustworthiness in deployment. This motivates the need for principled understanding and mitigation of video hallucination as a prerequisite for building dependable video understanding systems.

Prior surveys have established hallucination as a well-defined research problem for large language models (LLMs) in the text domain. Works such as (Huang et al., 2025b; Tonmoy et al., 2024) have provided principled taxonomies, analyzed failure mechanisms, and summarized mitigation strategies. There are also existing surveys on hallucination in large vision-language models (VLMs) for the image domain, including categorization of hallucination behaviors and representative mitigation methods (Liu et al., 2024a; Bai et al., 2024). However, extending these established perspectives to video understanding is non-trivial, as video introduces temporal and event-level challenges that lead to unique hallucination behaviors.

Video understanding goes beyond image-based perception: it requires temporal modeling, event-level reasoning, and long-range information integration across largely many frames. While video large language models (Vid-LLMs) certainly inherit certain perception-level hallucinations from image-based VLMs (e.g., object existence, attributes, and spatial relations), many failures are inherently video-specific. For example, models

may mishandle temporal understanding, misread motion and action dynamics, or lose information under long-context memory and retrieval even when single-frame perception is correct (Chang et al., 2025; Tang et al., 2025e). Video understanding frequently involves higher-level reasoning over events, including causality, intent, and multi-hop logic, where models may produce confident yet unsupported explanations (Jiang et al., 2025b).

Despite substantial recent efforts devoted to advancing Vid-LLMs, their hallucinations have not yet been systematically reviewed. Table 1 summarizes the coverage of related surveys, grouped into general video-understanding dimensions and hallucination-specific dimensions. The six video understanding surveys (Zhong et al., 2022; Madan et al., 2024; Nguyen et al., 2024; Zhou et al., 2024; Tang et al., 2025c; Zou et al., 2024) organize the literature around model architectures, training paradigms, datasets, and benchmark performance, characterizing what video models can do and how to improve task-level accuracy. As shown in the left block of Table 1, they provide broad coverage of models, tasks, benchmarks, and open challenges. Hallucination, however, is at most noted in passing: only Tang et al. (2025c) and Nguyen et al. (2024) mention it, and merely as a single open problem among many rather than a studied phenomenon. Across all six, failure cases are treated as general challenges rather than being formalized into distinct types with targeted remedies.

Two cross-modal hallucination surveys (Sahoo et al., 2024; Xia et al., 2025) do treat hallucination as a primary topic, but both apply taxonomies that are uniform across text, image, and video modalities. Neither defines hallucination in terms of video's unique properties—temporal dynamics, motion, and long-context dependencies—nor decomposes it into video-specific failure modes. Critically, neither organizes mitigation strategies by the type of hallucination being addressed. This survey fills this gap, as shown in the final row of Table 1.

| Survey | General Video-Understanding Coverage | | | Hallucination-Specific Coverage | | | | |
|---|---|---|---|---|---|---|---|---|
| | Models & Arch. | Tasks & Benchmarks | Challenges & Future | Hallu. Focus | Video-Spec. Definition | Hallu. Taxonomy | Hallu. Benchmarks | Mitigation by Hallu. Type |
| Zhong et al. (2022) | ✓ | ✓ | ✓ | ✗ | ✗ | ✗ | ✗ | ✗ |
| Madan et al. (2024) | ✓ | ✓ | ✓ | ✗ | ✗ | ✗ | ✗ | ✗ |
| Nguyen et al. (2024) | ✓ | ✓ | ✓ | ○ | ✗ | ✗ | ✗ | ✗ |
| Zhou et al. (2024) | ✓ | ✓ | ✓ | ✗ | ✗ | ✗ | ✗ | ✗ |
| Tang et al. (2025c) | ✓ | ✓ | ✓ | ○ | ✗ | ✗ | ✗ | ✗ |
| Zou et al. (2024) | ✓ | ✓ | ✓ | ✗ | ✗ | ✗ | ✗ | ✗ |
| Sahoo et al. (2024) | ○ | ○ | ✓ | ✓ | ✗ | ○ | ○ | ✗ |
| Xia et al. (2025) | ○ | ○ | ✓ | ✓ | ✗ | ○ | ○ | ✗ |
| This survey | ○ | ○ | ✓ | ✓ | ✓ | ✓ | ✓ | ✓ |

Table 1: Comparison of this survey with closely related prior work, grouped into *general video-understanding* coverage (left) and *hallucination-specific* coverage (right). **Legend:** ✓ = covered; ✗ = not covered; ○ = partially covered.

In this survey, we provide a unified, video-centric taxonomy of hallucination, organize evaluation benchmarks and mitigation techniques around it, analyze which mitigation foundations are most effective for each hallucination type, and identify key open challenges and future research directions. By structuring prior work in this way, the survey enables both researchers and practitioners to reason about video understanding failures in a principled manner and to develop mitigation approaches that address specific hallucination mechanisms and real-world deployment scenarios.

The contributions of this survey include:

- **Comprehensive taxonomy of video understanding hallucination.** We provide six major categories (Section 3): (1) temporal understanding hallucination, (2) motion and action dynamics hallucination, (3) long-context video understanding hallucination, (4) object hallucination, (5) causal and reasoning hallucination, and (6) cross-modal hallucination.

- **Benchmark suite organized by hallucination type.** We compile representative evaluation benchmarks and organize them under the six hallucination categories (Section 4), providing a structured, per-category reference for diagnosing and measuring each failure mode.

- **Systematic categorization of mitigation techniques and their effectiveness.** We first distill recent mitigation research into five foundational method families—reinforcement learning and preference optimization, supervision augmentation, structured representations, long-context information management, and inference-time intervention—organized by where they intervene in the Vid-LLM pipeline, and analyze how effectively each addresses every hallucination type (Section 5). Building on these foundations, we then review concrete mitigation methods organized by hallucination category (Section 6), aligning each technique with the specific failure mechanism it targets.

- **Forward-looking research directions toward trustworthy Vid-LLMs.**

  Beyond summarizing existing methods, we outline three promising directions for improving Vid-LLM reliability (Section 7): hallucination diagnosis and root-cause attribution frameworks that trace entangled, multi-symptom hallucinations back to their earliest failure mechanism along the Vid-LLM pipeline; out-of-schema strategies that enable models to handle novel entities and unseen event compositions and to recognize the boundaries of their own knowledge; and world-modeling approaches that ground temporal causality and action dynamics in explicit physical-world representations.

## 2 Hallucination in the Era of Video Understanding

### 2.1 Video Large Language Models

Vid-LLMs are multimodal foundation models that integrate video perception with LLMs to perform video understanding and reasoning tasks under natural language instructions. These tasks include video question answering, temporal grounding, video captioning, and causal inference (Tang et al., 2025c).

Vid-LLMs extend VLMs from static images to dynamic video inputs. While VLMs encode a single or a few images into visual tokens aligned with a language model, Vid-LLMs must process frame sequences or clips and model their temporal dependencies (Liu et al., 2024a).

Most Vid-LLMs consist of three components: (i) *a video encoder* that extracts spatio-temporal representations from video frames, (ii) *a temporal aggregation or memory mechanism* that compresses and organizes long video streams, and (iii) *a large language model* that performs reasoning and generation conditioned on the encoded video content (Nguyen et al., 2024).

Vid-LLMs face substantially higher complexity than image-based VLMs. Videos are orders of magnitude longer than images in both input size and semantic scope, often containing multiple events, evolving states, and long-range dependencies. Vid-LLMs must not only recognize objects and attributes, but also understand temporal order, motion patterns, event boundaries, and cross-event relations (Sahoo et al., 2024). These requirements fundamentally change both the failure modes and reliability properties of the model.

### 2.2 Hallucination in Vid-LLMs

In Vid-LLMs, hallucination refers to generated content that lacks faithful grounding in the actual video input. A hallucination can occur when a Vid-LLM produces descriptions, answers, or explanations that contradict or are unsupported by the visual-temporal evidence present in the video. It is important to distinguish hallucination from general prediction errors. In the broader video understanding literature, failures are often studied as perception, or localization problems. In this survey, we adopt a grounding-centric perspective and treat such failures as hallucinations. This enables a more systematic analysis of mitigation strategies, as methods that improve grounding fidelity can be studied under a unified framework regardless of whether the underlying failure originates from perception, localization, or memory.

This definition parallels hallucination in image-based large VLMs, where hallucination is defined as misalignment between image input and textual output (Liu et al., 2024a). Clearly, hallucination in Vid-LLMs can

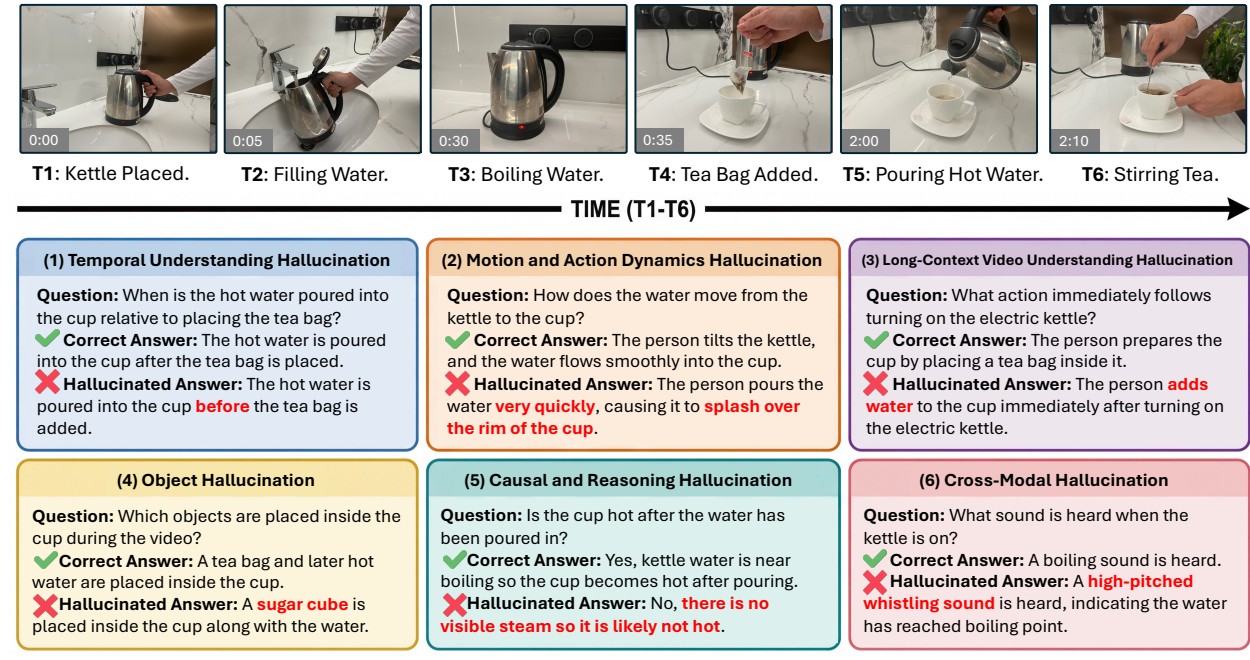

Figure 1: **Simple illustration of major video-understanding hallucination types.** Using a tea-making example with a timeline of key moments (T1–T6), the figure contrasts correct and hallucinated answers across six categories: (1) temporal understanding hallucination, (2) motion and action dynamics hallucination, (3) long-context video understanding hallucination, (4) object hallucination, (5) causal and reasoning hallucination, and (6) cross-modal hallucination.

exhibit fundamentally different characteristics due to the temporal and sequential nature of video data. This complexity arises from the need for Vid-LLMs to reason over multiple frames and, in many cases, across extended clips (Tang et al., 2025c).

In particular, a Vid-LLM may correctly recognize objects in individual frames yet hallucinate how these objects move, interact, or change over time. The model may invent actions that never occur, misorder events, attribute wrong causality, or confidently summarize events that are absent from the video.

Also, in long videos with minutes or even hours of visual content, models may forget earlier events, merge attributes from different temporal segments, or retrieve incorrect or incomplete memories when answering downstream queries. These distinctions indicate that hallucination in video understanding requires a dedicated formulation that explicitly accounts for temporal dynamics, motion, and long-context dependencies.

## 2.3 Literature Collection Methodology

To provide a comprehensive and systematic review of hallucination in video understanding, we conducted a structured survey of literature published between 2022 and 2026, covering the period during which Vid-LLMs and related video-language foundation models rapidly emerged. We chose 2022 as the starting year because it marks the transition away from the task-specific, discriminative paradigm that previously dominated video understanding—built around closed-set objectives such as action recognition, temporal localization, and video captioning—and toward the generative, instruction-following Vid-LLMs. Generative video-capable foundation models first appeared in 2022 (Alayrac et al., 2022), followed by a rapid wave of dedicated Vid-LLMs in 2023 (Li et al., 2023b; Zhang et al., 2023b; Maaz et al., 2023). This inflection is also reflected empirically in our corpus: the number of hallucination-related papers grows from roughly one hundred in 2022 to more than five times that figure by 2025 (Appendix A), confirming 2022 as a natural and conservative starting point that captures the field essentially from its inception. Candidate papers were identified through keyword-based searches and citation analysis of relevant works. Beyond these primary papers from 2022–

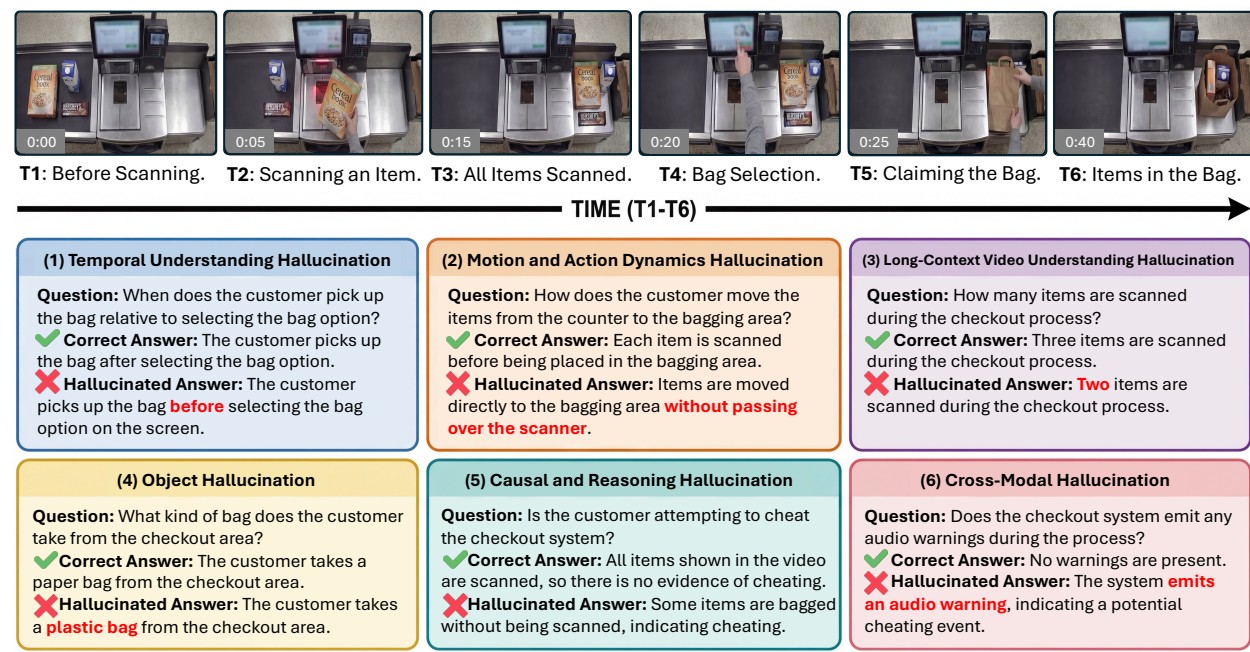

Figure 2: **Real-world example of major video-understanding hallucination types.** Using a grocery store self-checking example with a timeline of key moments (T1–T6), the figure contrasts correct and hallucinated answers across six categories: (1) temporal understanding hallucination, (2) motion and action dynamics hallucination, (3) long-context video understanding hallucination, (4) object hallucination, (5) causal and reasoning hallucination, and (6) cross-modal hallucination.

2026, we additionally collected the benchmarks and datasets from early years that these works rely on for evaluation.

We included papers that (1) analyze hallucination phenomena in video understanding systems, (2) introduce hallucination-related benchmarks, datasets, or evaluation protocols, (3) propose mitigation techniques for hallucination and grounding failures in Vid-LLMs, or (4) address reliability challenges closely related to hallucination, such as temporal grounding, long-context reasoning, and multimodal alignment. We excluded papers focusing exclusively on image-only hallucination without video-specific insights, works unrelated to reliability or grounding in video understanding, and duplicate versions of the same work. The collected literature was subsequently reviewed, selected and organized according to the hallucination taxonomy and mitigation framework proposed in this survey.

## 3 Hallucination Taxonomy

To systematically analyze Vid-LLMs' hallucination, we organize existing failure modes into a unified taxonomy. Figure 3 groups hallucination phenomena into six major categories. Each category captures distinct reliability failures that can arise while other aspects of video perception remain accurate.

Figures 1 and 2 demonstrate different hallucination categories through concrete examples.

### 3.1 Temporal Understanding Hallucination

Temporal understanding hallucination is the failure where a Vid-LLM produces incorrect temporal ordering or event timing. The model may correctly recognize which events are present but misrepresent their chronological order, relative timing, or durations. Prior work on temporal reasoning and moment-level understanding has shown that such temporal errors remain prevalent (Ahn et al., 2025; Huang et al., 2024a). We identify two major manifestations of this category: *sequencing errors* and *temporal grounding errors.*

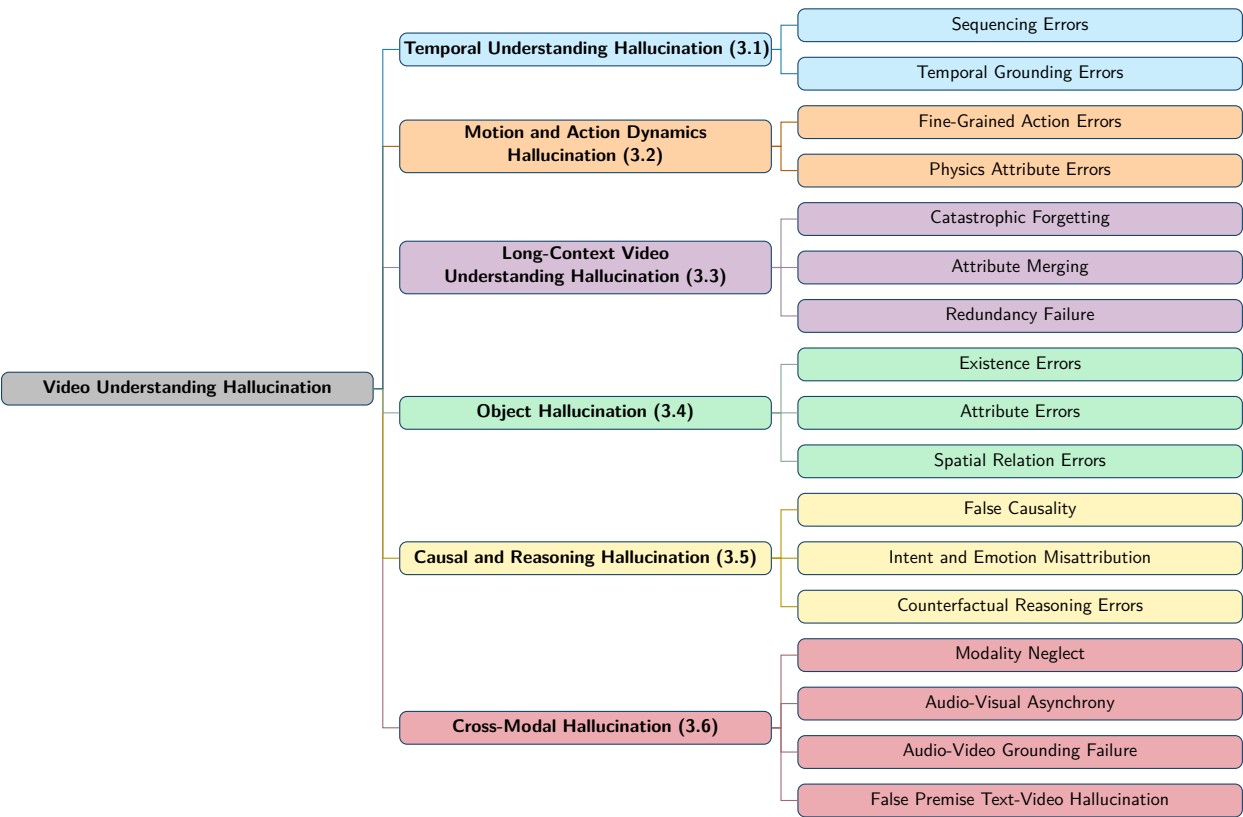

Figure 3: Taxonomy of video understanding hallucination, organized by category and error type.

**Sequencing Errors.** Sequencing errors arise when a model messes up the chronological order of events while correctly identifying the individual events themselves (Wu et al., 2025a; Plizzari et al., 2025; Li et al., 2025a). A common example is "the person sits down and then stands up" when the video actually shows standing and then sitting. Another case is "the player celebrates and then scores" when the score clearly precedes the celebration.

Empirical analyses of temporal ordering have shown that models frequently struggle with maintaining consistent event order, leading to swapped or reversed narratives in generated descriptions (Ahn et al., 2025; Xue et al., 2025b).

**Temporal Grounding Errors.** Temporal grounding errors arise *when* an event is claimed to happen along the timeline but assigned incorrect timestamps or boundaries. For instance, stating that "a person starts running" at 0:10 when it actually starts at another time, or describing that "the accident happens in the last few seconds" when it occurs near the middle of the video (Guo et al., 2025c; Chen et al., 2024d). Another common symptom is over-extending event duration, where a brief hand gesture is described as spanning an entire segment. These errors have been widely documented in fine-grained grounding and moment localization tasks (Huang et al., 2024a; Wang et al., 2024a; Wu et al., 2025e; Pramanick et al., 2025).

## 3.2 Motion and Action Dynamics Hallucination

Motion and action dynamics hallucination refers to failures in capturing how movements unfold over time, which is important when to distinguishing between actions relies on subtle temporal cues or when correct understanding requires consistent modeling of physical trajectories. Such hallucinations arise when models fail to encode fine-grained inter-frame changes, causing them to hallucinate motion that never occurred or

misinterpret visually similar actions. We highlight two recurring error patterns: *fine-grained action errors* and *physics attribute errors.*

**Fine-Grained Action Errors.** Even when the model understands the scene, it can fail to distinguish subtle motions (Du et al., 2025; Hong et al., 2025). For example, describing a person as running when they are actually walking quickly, or claiming that a person is rubbing their eyes when they are merely adjusting their glasses. The literature attributes it to insufficient sensitivity to small-amplitude temporal differences across adjacent frames (Shao et al., 2020; Liu et al., 2022b).

**Physics Attribute Errors.** Physics attribute errors are failures in tracking the kinematic evolution of objects (Wu et al., 2025b). Typical mistakes include stating that a ball rolls to the left when it does to the right, or stating the speed wrong. Models may also hallucinate physically implausible transitions, such as sudden direction changes without a transition. Empirical studies such as PhyVLLM (Zhan et al., 2025) and PiTe (Liu et al., 2024d) claim that incorrect reasoning about motion direction, trajectory, and velocity remains common in Vid-LLMs.

## 3.3 Long-Context Video Understanding Hallucination

Long-context video understanding hallucination arises as its name suggests. When models integrate evidence across thousands to millions of frames, constraints from limited context windows, compressed representations, sliding windows, or limited external memory can distort what the model recalls or attends to over time. We identify three characteristic error patterns: *catastrophic forgetting*, *attribute merging*, and *redundancy failure.*

**Catastrophic Forgetting.** Earlier evidence tends to gradually disappear as the model processes later frames (Xiong et al., 2025; Wang et al., 2024m). Catastrophic forgetting occurs when a model correctly had encoded an early event or fact but later fails to recall it or produces contradictory statements (Jin et al., 2025c; Yang et al., 2025c; Li et al., 2024d; Yuan et al., 2025). Consider an hour-long lecture video where a model correctly recognizes that the instructor taught something in the first few minutes, yet later claims it didn't happen when queried near the end.

**Attribute Merging.** Attribute merging refers to information *conflation*, i.e., the model incorrectly blends attributes of objects, people, or scenes that appear at different times (Lu et al., 2025). For example, after observing a red car early and a blue car later, the model incorrectly answers with a red car in the later scene.

**Redundancy Failure.** Long videos often contain large volumes of repetitive or low-information frames, which can suppress attention to sparse but critical evidence (Yunzhuzhang et al., 2025; Hu et al., 2025a; Yu et al., 2024a). Redundancy failure arises when models become overwhelmed by static scenes or slow-changing environments, causing salient events to get diluted, thus triggering hallucinated details (Zhang et al., 2025f;b; Tan et al., 2026). In an hour-long security camera feed with mostly event-free frames, a person may briefly walk through the hallway mid-video, yet the model fails to identify or report this event due to the dominance of redundant low-activity content. The signal-to-noise ratio becomes particularly problematic in such scenarios.

## 3.4 Object Hallucination

Object hallucination describes perception-level failures where a model cannot reliably extract object-centric evidence and consequently generates unsupported or inconsistent claims. Compared to static images, object understanding in videos presents greater challenges, as objects may undergo motion, occlusion, viewpoint changes, and appearance variation. Hallucination may arise in identifying *what* objects exist, *which* attributes they exhibit, or *how* they are spatially arranged in a dynamic video setting. We summarize three representative forms of hallucination: *existence errors*, *attribute errors*, and *spatial relation errors.*

**Existence Errors.** Existence errors are hallucinated objects that do not appear in the video. Temporal sparsity and occlusion often exacerbate such failures, where objects may be visible only briefly or partially,

making consistent presence tracking difficult (Lusha et al., 2025; Rawal et al., 2025; Wang et al., 2024l). Consider a cooking video that shows only a person stirring a pot yet a model may hallucinate the presence of a knife even though no knife ever appears. This type of unsupported entity generation has been reported in (Chang et al., 2025; Wang et al., 2025a).

**Attribute Errors.** Attribute errors refer to misidentified visual properties, such as color, shape, size, or textual content, often due to lighting variation, motion blur, and viewpoint shifts (Rawal et al., 2025; Patraucean et al., 2023; Yang et al., 2025a). The model may correctly recognize a car but incorrectly claim it is red when it is actually blue. Such errors frequently occur in long-video understanding, where incomplete or compressed representations amplify the model's confusion (Wang et al., 2024g; Zou et al., 2024).

**Spatial Relation Errors.** Spatial relation errors are incorrectly described configurations or relative positions of objects. For example, it may state "the cup is on the table" when the video shows the cup being held in a hand. These errors may be attributed to the changing spatial groundings like object motion and camera movement (Wang et al., 2024l). Imperfect spatial grounding may cause models to generate plausible but unsupported relations (Wang et al., 2025c; Ouyang et al., 2025a; Tang et al., 2025c).

## 3.5 Causal and Reasoning Hallucination

Causal and reasoning hallucination arises when a model produces logically unsupported conclusions on top of otherwise correct perception. Here, models hallucinate unobservable concepts such as motivations, emotions, intentions, or causal relations. Unlike object hallucination (e.g., hallucinating an object), this category reflects breakdowns in higher-level inference over events and interactions (Jiang et al., 2025b). Such failures are particularly salient in video understanding because correct reasoning often requires integrating temporal cues, social context; and multi-step event structures. We identify three representative forms: *false causality*, *intent and emotion misattribution*, and *counterfactual reasoning errors*.

**False Causality.** False causality is a spurious cause-to-effect link: the model correctly recognizes events but constructs an unsupported causal explanation often by relying on stereotypical correlations instead of video evidence (Guda et al., 2024; Li et al., 2022b; Chen et al., 2024e). For example, seeing a video where a man suddenly starts running, the model may claim he is running because someone is chasing him. Models often conflate correlation with causation and over-rely on commonsense priors (Liu et al., 2023c; Wei et al., 2023).

**Intent and Emotion Misattribution.** Intent and emotion misattribution is the *over-interpretation of social evidence*: the model maps observable behavior into mental-state labels without sufficient visual support (Li et al., 2023a; Ye et al., 2024; Peng et al., 2025; Chen et al., 2025k; Kong et al., 2025). For instance, a model may claim that "two people are arguing" based on their gestures, when the video actually shows them laughing with gestures.

**Counterfactual Reasoning Errors.** Counterfactual reasoning errors emerge under *hypothetical intervention* queries, where correctness requires reasoning about alternative outcomes consistent with rules and causal structure (Chen et al., 2025h; Huang et al., 2025d; Poppi et al., 2026) indicated by the video. For example, when asked "What would happen if the player missed the shot?", the model may hallucinate an outcome that contradicts the rules of the game played in the video.

## 3.6 Cross-Modal Hallucination

Cross-modal hallucination arises when models incorrectly integrate multimodal evidence (e.g., vision, audio, and text), producing outputs that violate cross-modal consistency (Guo et al., 2025d; Shen et al., 2023). We categorize it into four recurring patterns: *modality neglect*, *audio-visual asynchrony*, *audio-video grounding failure*; and *false premise text–video hallucination*.

**Modality Neglect.** Modality neglect occurs when one modality (typically vision) dominates, causing the model to ignore another (typically audio), even when it is salient (Guo et al., 2025d; Shen et al., 2023; Li et al., 2025k). For example, a video shows a person smiling and nodding while the audio contains another person saying "I feel terrible and I need help", but a model that discards the audio clues may output "a person is listening and agreeing" (Hanna-Asaad et al., 2024; Cheng et al., 2024).

**Audio-Visual Asynchrony.** Audio-visual asynchrony is a temporal alignment failure: the model uses both modalities but binds auditory evidence to the wrong visual moment (Goel et al., 2024; Tang et al., 2025d). For instance, it may describe a door slamming sound as occurring when a person merely approaches the door, rather than when the door actually closes.

**Audio-Video Grounding Failure.** Audio-video grounding failure concerns *source attribution*: the model assigns an audio signal to an incorrect visual entity or invents a source not responsible for the sound (Li et al., 2022a; Jiang & Yin, 2023). For example, in a scene with multiple people on screen and background television audio, the model may hallucinate that one of the visible individuals is speaking, rather than the off-screen TV (Li et al., 2024e).

**False Premise Text-Video Hallucination.** False premise text-video hallucination arises when the textual input embeds an assumption that conflicts with the video, and the model prioritizes linguistic grounding over visual grounding (Gao et al., 2025b). For example, when asked about the color of a cat in a video that contains no cat, the model may answer "the cat is white," hallucinating to satisfy the textual premise (Seth et al., 2025).

## 4 Benchmarks and Evaluation

Evaluating each hallucination type is essential for understanding model reliability. With robust benchmarks, we can quantify the extent to which a model suffers from a specific hallucination type and design targeted mitigation techniques accordingly. In this section, we review evaluation benchmarks organized according to the hallucination taxonomy introduced in Section 3, focusing on the benchmarks most widely used to evaluate the mitigation techniques discussed in Section 6. Table 2 provides a comprehensive overview.

### 4.1 Temporal Understanding Hallucination Benchmarks

Charades-STA (Gao et al., 2017) is the most widely used benchmark for temporal sentence grounding, with over six thousand videos and sixteen thousand query-segment pairs. It measures IoU-based metrics (R1@0.5, R1@0.7) that directly quantify temporal localization accuracy. Methods such as Time-R1 (Wang et al., 2025j) and DisTime (Zeng et al., 2025c) use Charades-STA as their primary evaluation benchmark, making it a standard reference point for comparing temporal grounding approaches.

TempCompass (Liu et al., 2024e) is one of the earliest benchmarks designed to directly probe temporal perception in Vid-LLMs. It evaluates five temporal aspects—action, speed, direction, attribute change, and event order—through a multi-format design including multiple-choice, yes/no, caption matching, and caption generation. A key methodological contribution is its conflicting video construction: reversing, spatial concatenation, and temporal concatenation create paired videos where single-frame shortcuts yield correct answers for one video but incorrect answers for its counterpart, preventing models from exploiting static visual cues or language priors.

ActivityNet-RTL (Huang et al., 2024b) assesses temporal reasoning and localization over diverse real-world activities, serving as a complementary grounding benchmark alongside Charades-STA. From a diagnostic perspective, EventHallusion (Zhang et al., 2024b) diagnoses temporal event hallucinations by disentangling two sources: susceptibility to language priors that suggest incorrect event sequences, and susceptibility to vision-language correlation biases that distort temporal event interpretation. Taking a complementary approach, Charades-CON (Jung et al., 2025) reformulates temporal grounding as a verification task, testing whether timestamp encoding genuinely supports temporal comprehension.

Table 2: Summary of benchmarks for evaluating video hallucination across the six hallucination categories. Each benchmark is listed under its primary category. Abbreviations: MC = multiple-choice; QA = question answering; IoU = Intersection-over-Union; ST = spatio-temporal; box = bounding box.

| Hallucination Category | Benchmark | Year | Eval Format | Key Evaluation Focus |
|---|---|---|---|---|
| **Temporal Understanding** | Charades-STA (Gao et al., 2017) | 2017 | Grounding (IoU) | Temporal sentence grounding |
| | TempCompass (Liu et al., 2024e) | 2024 | Multi-format | Temporal perception across 5 aspects |
| | EventHallusion (Zhang et al., 2024b) | 2024 | MC QA | Temporal event hallucination diagnosis |
| | Charades-CON (Jung et al., 2025) | 2025 | Verification | Temporal consistency verification |
| | ActivityNet-RTL (Huang et al., 2024b) | 2024 | Grounding (IoU) | Temporal reasoning and localization |
| **Motion and Action Dynamics** | MVBench (Li et al., 2024a) | 2024 | MC QA | Motion-aware video understanding |
| | MotionBench (Hong et al., 2025) | 2025 | MC QA | Fine-grained motion understanding |
| | Perception-Test (Patraucean et al., 2023) | 2023 | MC / Open QA | Action and motion perception diagnostics |
| | ActivityNet-QA (Yu et al., 2019) | 2019 | Open QA | Activity-centric video QA |
| | UNSCENE (Bae et al., 2025) | 2025 | MC QA | Entangled motion hallucination |
| **Long-Context Video Understanding** | Video-MME (Fu et al., 2025) | 2024 | MC QA | Duration-dependent video comprehension |
| | MLVU (Zhou et al., 2025b) | 2025 | MC / Open QA | Multi-task long video understanding |
| | NIAVH (Wang et al., 2024m) | 2024 | Open QA | Needle-in-a-video-haystack |
| | LVBench (Wang et al., 2025g) | 2025 | MC QA | Extreme long video understanding |
| | ELV-Halluc (Lu et al., 2025) | 2025 | Multi-format | Long-form semantic aggregation hallucination |
| **Object Hallucination** | VideoHallucer (Wang et al., 2024l) | 2024 | Binary QA | Intrinsic and extrinsic object hallucination |
| | VidSTG (Zhang et al., 2020) | 2020 | Grounding (box) | Object localization via ST grounding |
| | HCSTVG (Tang et al., 2021) | 2021 | Grounding (box) | Person localization via ST grounding |
| | CLEVRER (Yi et al., 2019) | 2019 | MC QA | Object dynamics and interaction reasoning |
| | TGIF-QA (Jang et al., 2017) | 2017 | MC / Open QA | Object state and action transition QA |
| **Causal and Reasoning** | NExT-QA (Xiao et al., 2021) | 2021 | MC QA | Causal and sequential action reasoning |
| | TrafficQA (Xu et al., 2021) | 2021 | MC QA | Traffic event causal reasoning |
| | AGQA (Grunde-McLaughlin et al., 2021) | 2021 | MC QA | Compositional spatio-temporal reasoning |
| | CounterVQA (Chen et al., 2025h) | 2025 | MC QA | Counterfactual reasoning |
| | CounterVid (Poppi et al., 2026) | 2026 | Preference | Counterfactual video discrimination |
| | ViRectify (Hei et al., 2025) | 2025 | Correction | Video reasoning error correction |
| | MMR-V (Zhu et al., 2025a) | 2025 | Open QA | Multi-step grounded reasoning |
| **Cross-Modal** | MUSIC-AVQA (Li et al., 2022a) | 2022 | MC QA | Audio-visual question answering |
| | VELOCITI (Saravanan et al., 2025) | 2025 | Entailment | Compositional video-language entailment |
| | Video-CSR (Liu et al., 2023b) | 2023 | Captioning | Cross-modal video digest creation |

## 4.2 Motion and Action Dynamics Hallucination Benchmarks

MVBench (Li et al., 2024a) includes motion-specific subtasks (action sequence, action prediction, unexpected action) among its 20 evaluation dimensions. Its broad coverage makes it the most widely adopted benchmark for evaluating motion understanding in the mitigation literature, used by methods ranging from object-centric tracking (Feng et al., 2024) to specialized motion architectures (Rasekh et al., 2025) and dynamic token manipulation (Liu et al., 2024c).

MotionBench (Hong et al., 2025) is the first benchmark specifically designed to probe fine-grained video motion understanding in Vid-LLMs. It organizes questions across motion-specific subtasks, including camera motion, object motion, and motion-dependent attributes. MotionBench revealed that even state-of-the-art models struggle significantly with fine-grained motion tasks, particularly when distinguishing camera ego-motion from object motion.

Perception-Test (Patraucean et al., 2023) provides a diagnostic benchmark with evaluation of skills including object tracking, point tracking, and temporal action detection, each requiring fine-grained motion sensitivity. For real-world scenarios, ActivityNet-QA (Yu et al., 2019) provides large-scale video question answering over diverse real-world activities, requiring models to recognize and reason about action dynamics in complex, untrimmed videos. More recently, UNSCENE (Bae et al., 2025) evaluates motion hallucinations arising from entangled causal attention within multimodal models.

### 4.3 Long-Context Video Understanding Hallucination Benchmarks

Video-MME (Fu et al., 2025) is the most widely used benchmark for long-context evaluation, spanning short (less than 2 minutes), medium (4–15 minutes), and long (30–60 minutes) video durations. This multi-duration design enables systematic analysis of how hallucination rates change with video length. Token compression and retrieval-augmented generation (RAG)-based mitigation methods rely heavily on Video-MME as their primary evaluation benchmark.

MLVU (Zhou et al., 2025b) is a multi-task long video understanding benchmark covering nine task categories including topic reasoning, anomaly recognition, and needle QA, with videos spanning several minutes to over two hours. RAG-based approaches such as Vgent (Shen et al., 2025) and MemVid (Yuan et al., 2025) use MLVU as a primary evaluation benchmark for retrieval quality.

NIAVH (Wang et al., 2024m) adapts the needle-in-a-haystack paradigm to video, embedding a target event within a long video that models must locate and reason about despite extensive surrounding content. This directly evaluates redundancy failure, where salient events are diluted by low-information frames. Pushing the duration further, LVBench (Wang et al., 2025g) specifically tests extreme long video understanding with videos averaging approximately one hour each. Beyond retrieval failures, ELV-Halluc (Lu et al., 2025) is specifically designed to benchmark semantic aggregation hallucinations in long video understanding, evaluating whether models incorrectly blend or conflate information from different temporal segments.

### 4.4 Object Hallucination Benchmarks

VideoHallucer (Wang et al., 2024l) is the first comprehensive benchmark dedicated to evaluating object hallucination in video. It evaluates *intrinsic* hallucinations (fabricated object relations, incorrect object attributes, and erroneous spatial details) and *extrinsic* hallucinations (objects or properties not present in the video). Its adversarial binary QA methodology pairs each basic yes/no question with a hallucinated counterpart, requiring models to answer both correctly to receive credit, distinguishing genuine object comprehension from lucky guessing.

VidSTG (Zhang et al., 2020) and HCSTVG (Tang et al., 2021) test spatio-temporal object grounding, measuring whether models can locate specific objects at specific times with bounding box precision. These benchmarks serve as the primary evaluation for object-centric tracking and grounding methods in Section 6.4.

CLEVRER (Yi et al., 2019) provides a controlled synthetic environment involving collision events, enabling precise evaluation of object existence and interaction reasoning. For real-world content, TGIF-QA (Jang et al., 2017) offers visual question answering on animated clips, widely used for evaluating object-level video QA through both multiple-choice and open-ended formats.

### 4.5 Causal and Reasoning Hallucination Benchmarks

NExT-QA (Xiao et al., 2021) emphasizes causal ("why") and sequential ("how") questions that require reasoning about action motivations and consequences. Its causal question design makes it a natural testbed for evaluating whether mitigation techniques genuinely improve causal reasoning or merely reduce surface-level errors.

TrafficQA (Xu et al., 2021) specifically targets causal reasoning about traffic events, including accident attribution and counterfactual reasoning about traffic scenarios. Moving to counterfactual reasoning, CounterVQA (Chen et al., 2025h) directly evaluates counterfactual reasoning at three progressive difficulty levels: Adjacent Counterfactual Inference (single-step), Long-Chain Counterfactual Inference (multi-hop), and Counterfactual Inference with Non-existent Events, enabling fine-grained diagnosis of where causal reasoning breaks down. CounterVid (Poppi et al., 2026) complements CounterVQA with a counterfactual video generation framework for preference-based evaluation. From a different angle, ViRectify (Hei et al., 2025) evaluates video reasoning correction by presenting models with flawed reasoning traces. MMR-V (Zhu et al., 2025a) benchmarks multi-step inference chains where each step must be grounded in visual evidence. At a larger scale, AGQA (Grunde-McLaughlin et al., 2021) provides compositional spatio-temporal reasoning with a large-scale machine-generated QA corpus.

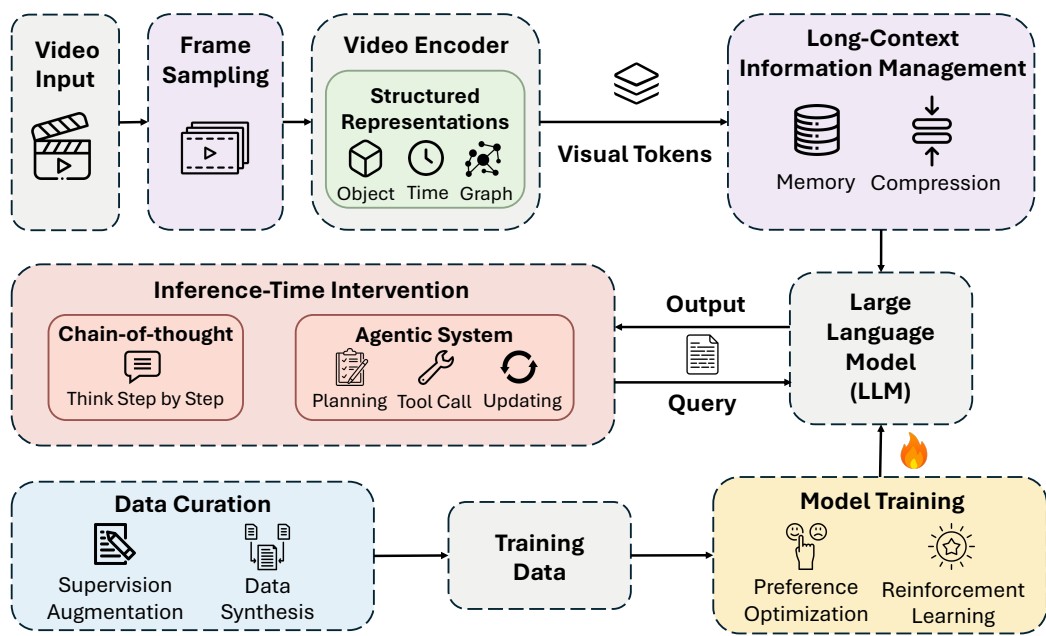

Figure 4: Illustration of representative hallucination mitigation strategies in Vid-LLMs organized by the stage of the pipeline where they intervene. Colored components highlight different methodological directions discussed in Section 5, with each color corresponding to one direction, while the grey blocks depict the standard Vid-LLMs pipeline.

### 4.6 Cross-Modal Hallucination Benchmarks

MUSIC-AVQA (Li et al., 2022a) is a large-scale audio-visual QA benchmark designed to evaluate audio-visual understanding in musical performance contexts. It requires models to jointly reason about what they see and hear, making it a natural testbed for modality neglect and audio-visual grounding failures.

VELOCITI (Saravanan et al., 2025) benchmarks video-language compositional reasoning with strict entailment, requiring that model outputs be logically entailed by the multimodal evidence rather than merely plausible. This design is effective for detecting cross-modal hallucinations where models generate linguistically fluent but evidentially unsupported claims. Complementing this, Video-CSR (Liu et al., 2023b) evaluates complex video digest creation across visual and linguistic modalities, testing whether models maintain cross-modal consistency in their generated summaries. The limited number of dedicated cross-modal hallucination benchmarks, compared to other categories, highlights this as an area where evaluation methodology has not yet kept pace with mitigation research.

## 5 Foundations of Hallucination Mitigation

Before summarising specific mitigation techniques in Section 6 below, we review recurring methods underlying recent hallucination mitigation research. As hallucination arises from failures at different stages of the Vid-LLM pipeline, existing mitigation approaches can be understood as interventions at the corresponding stages. The following subsections review these methods, including reinforcement learning (RL) and preference optimization, supervision augmentation with synthetic data, structured representations, long-context video evidence management, and inference-time interventions, organized by where they intervene (see Figure 4). We then consolidate these families into a cross-cutting view of how strongly each addresses every hallucination category.

### 5.1 Reinforcement Learning and Preference Optimization

RL and preference optimization align LLMs with desired behaviors by introducing training objectives beyond standard likelihood-based learning (Zhang et al., 2025e; Rafailov et al., 2023). The language model is treated as a policy, and an external training signal evaluates generated outputs and updates the model parameters so that preferred responses become more likely (Liu et al., 2025c).

In RL-based methods, the model generates candidate outputs and receives reward signals measuring factual correctness, reasoning quality, or human preference alignment. Policy optimization then updates the model to reinforce higher-reward responses and suppress lower-reward ones. For Vid-LLMs, effective reward design relies on verifiable supervision tied to video-specific structure, such as timestamps, segment boundaries, object trajectories, and event sequences, rather than generic answer correctness alone (Feng et al., 2025; Chen, 2025).

Preference optimization instead trains on pairs of candidate responses where one is preferred over the other (Huang et al., 2025a). In video-language settings, many hallucinations arise from subtle temporal or spatial misalignments rather than completely wrong answers, so preference construction benefits from fine-grained temporal or motion-aware supervision rather than relying on coarse human judgments (Li et al., 2025f; Kulkarni & Fazli, 2025).

### 5.2 Supervision Augmentation and Synthetic Data

Standard video datasets emphasize coarse descriptions and contain little supervision for subtle temporal relations, object interactions, or causal reasoning. Models may therefore never encounter training examples that explicitly penalize hallucinations of these kinds.

Synthetic supervision addresses this by constructing additional training examples that highlight hallucination-prone scenarios. Synthetic pipelines perturb or recompose video-text pairs to train the model on correct evidence grounding, for instance, by shuffling or reversing event sequences (Chen et al., 2023b; Vani et al., 2025), and concatenating shorter clips to simulate long videos (Ren et al., 2024b).

However, synthetic supervision introduces a trade-off between controllability and realism: artificially constructed examples may not reflect natural video distributions. The additional fine-grained temporal signals also increase memory and token requirements, so the effectiveness of synthetic data depends on both the quality of generated supervision and whether the model architecture can preserve these signals during long-context processing.

### 5.3 Structured Representations

Structured representations modify models to keep entities, relations, and interactions explicit during reasoning, rather than mapping visual inputs directly into dense token embeddings. Such models may benefit from intermediate, interpretable structures like relational graphs, modular feature groups, and entity-based representations (Mao et al., 2022).

In video understanding, hallucinations can be attributed to planning errors, not only perception errors, for example, answering before it has localized the relevant moments or decomposed the reasoning path. Structured representations help because they force intermediate grounding decisions before the final answer. An effective design to address the issue links observations across multiple frames into temporally coherent units through temporal linking, trajectory aggregation, or timestamp-aware embeddings; they encode both appearance features and temporal indices (Guo et al., 2025c; Wang et al., 2024a).

Another approach organizes video information into scene graphs or dynamic interaction graphs, representing entities as nodes and their spatial or temporal interactions as edges (Du & Wang, 2025). Reasoning over explicit relationships helps prevent the model from introducing unsupported objects, relations, or interactions (Huang et al., 2025c; Chu et al., 2025).

In practice, structured representations alone are insufficient. Their effectiveness depends on how the model uses them, which require trainings or agent pipelines that focus attention on query-relevant relations and evidence (Malik et al., 2025).

## 5.4 Long-Context Information Management

Long videos introduce a fundamental challenge: a single video may contain thousands of frames, while the model can process only a limited number of visual tokens (Xu et al., 2025c). Models must therefore selectively retain, compress, and retrieve information from earlier parts of the video.

Memory-based approaches augment the model with external or persistent memory that stores compact representations of previously processed segments, such as learned embeddings, key-value representations, and segment-level summaries. At query time, relevant memory entries are retrieved through similarity matching (Song et al., 2023; Jin et al., 2025c).

Token compression methods reduce visual tokens by merging redundant tokens using clustering, similarity-based merging, or learned compressors, fitting longer videos within fixed context budgets (Weng et al., 2024; Guo et al., 2025b). Empirically, effective compression preserves the spatial and temporal structure of events rather than minimizing token counts uniformly, because collapsing distinct moments into indistinguishable tokens can lead to hallucinated actions or causal relations (Sun et al., 2025a).

Hallucination in long videos can also be a retrieval failure disguised as a reasoning failure (Yu et al., 2026). Frame selection methods estimate frame relevance using motion cues, saliency scores, or query-conditioned scoring and select informative subsets for reasoning (Tang et al., 2025b; Guo et al., 2025a). Retrieval-based pipelines index video segments externally and retrieve the most relevant clips for the query (Ataallah et al., 2024; Yuan et al., 2025).

## 5.5 Inference-Time Intervention

Inference-time intervention methods work during generation instead of training, which has been popular in recent years in all LLM modalities. Chain-of-thought (CoT) prompting, for example, inserts reasoning demonstrations or instructions such as "think step by step" to structure how the model produces tokens (Wei et al., 2022; Zhang et al., 2022).

For Vid-LLMs, the prompt encourages intermediate reasoning steps that reference specific video observations, e.g., events, object interactions, and temporal relations, before integrating them into a final conclusion (Zhang et al., 2025i; Jin et al., 2025b).

Beyond linear reasoning traces, agentic reasoning procedures organize inference as an iterative loop of planning, tool calling, and result updating (Luo et al., 2025a; Huang et al., 2024c; Ishibashi & Nishimura, 2024). In video understanding, as we mentioned before, hallucination often arises from failures in summarization and planning rather than perception. Agentic frameworks address this by first planning which parts of the video to inspect, then performing targeted operations such as selecting relevant frames or retrieving supporting segments (Zhang et al., 2025c; Chen et al., 2025a). The final answer is produced only after sufficient visual evidence has been collected.

## 5.6 Effectiveness on Different Hallucination Types

The five method families above are not applied uniformly across hallucination types; each family targets the failure stages where it is most effective. Table 3 summarizes this correspondence, previewing the per-category discussion of Section 6. Each cell rates how much a family contributes to mitigating that hallucination type: ✓✓✓ (strong), ✓✓ (moderate), and ✓ (limited) reflect decreasing levels of contribution, while ✗ denotes a minor or negligible contribution. Each cell summarizes the systematic per-category analysis and quality check of the surveyed methods. For every method family and hallucination type, we examine both how prominently that family is used and the effectiveness it reports on the relevant benchmarks.

Table 3: Effectiveness of the five mitigation foundations (Section 5) on the six hallucination categories (Section 3). ✓✓✓ / ✓✓ / ✓ denote a strong, moderate, and limited contribution to mitigating the corresponding hallucination type, respectively, judged by prominence in the mitigation literature and reported effectiveness; ✗ indicates that the family makes only a minor or negligible contribution to that category.

| Mitigation Foundation | Temporal | Motion | Long-Context | Object | Causal & Reasoning | Cross-Modal |
|---|---|---|---|---|---|---|
| RL & Preference Optimization | ✓✓✓ | ✓✓ | ✗ | ✓✓ | ✓✓✓ | ✗ |
| Supervision Augmentation & Synthetic Data | ✓✓ | ✗ | ✗ | ✗ | ✗ | ✗ |
| Structured Representations | ✓✓✓ | ✓✓✓ | ✗ | ✓✓✓ | ✓✓ | ✓✓✓ |
| Long-Context Information Management | ✗ | ✗ | ✓✓✓ | ✗ | ✗ | ✗ |
| Inference-Time Intervention | ✓ | ✓✓ | ✓✓ | ✓✓ | ✓✓✓ | ✗ |

Several patterns emerge from this view. First, *structured representations* are the most pervasive remedy: explicit time tokens and time-aware encoders for temporal grounding, object-centric trajectories and graph-based reasoning for motion and object hallucination, and rotary time embeddings or interleaved encoding for cross-modal grounding all impose intermediate structure that anchors generation to evidence. Second, *RL and preference optimization* is broadly applicable and especially dominant for temporal and causal-reasoning hallucinations, where verifiable rewards directly penalize ungrounded outputs. Third, *long-context information management* is, by construction, specialized to long-context hallucination and is rarely repurposed for other categories. Fourth, *synthetic and augmented supervision* contributes substantially only to temporal hallucination, where order- and progression-perturbed data directly counteract shortcut learning, while playing at most a marginal role elsewhere. Finally, *inference-time intervention* contributes most to causal and reasoning hallucination through chain-of-thought and multi-agent verification, while offering lighter-weight, training-free gains for motion, long-context, and object hallucinations.

A clear gap is visible in the cross-modal column: it is addressed almost exclusively through representation-level temporal synchronization and contrastive alignment, with essentially no use of RL, synthetic supervision, or inference-time intervention. This relatively underexplored intersection echoes the scarcity of dedicated cross-modal hallucination benchmarks noted in Section 4.6.

## 6 Mitigation Techniques

We now review concrete mitigation techniques organized according to the hallucination categories from Section 3, examining how different methods improve model faithfulness and reliability (see Figure 5).

### 6.1 Temporal Understanding Hallucination Mitigation

Temporal understanding hallucination is a major failure mode of Vid-LLMs, motivating the inventions of techniques that target temporal ordering, localization, and duration grounding. Existing approaches fall into three technical paradigms: (i) *preference optimization and reinforcement learning* with temporally grounded reward signals; (ii) *specialized architectures and inference-time interventions* that encode or diagnose temporal structure; (iii) *synthetic data generation and spatiotemporal augmentation* to expose models to temporally perturbed supervision (see Figure 6).

**Preference Optimization and Reinforcement Learning**    Early supervised fine-tuning approaches such as TimeChat (Ren et al., 2024a) and TRACE (Guo et al., 2024) penalize false negatives under autoregressive losses, causing overfitting and poor generalization. RL-based alternatives are motivated by the insight that temporal localization quality is better captured by task-specific reward signals than by token-level cross-entropy.

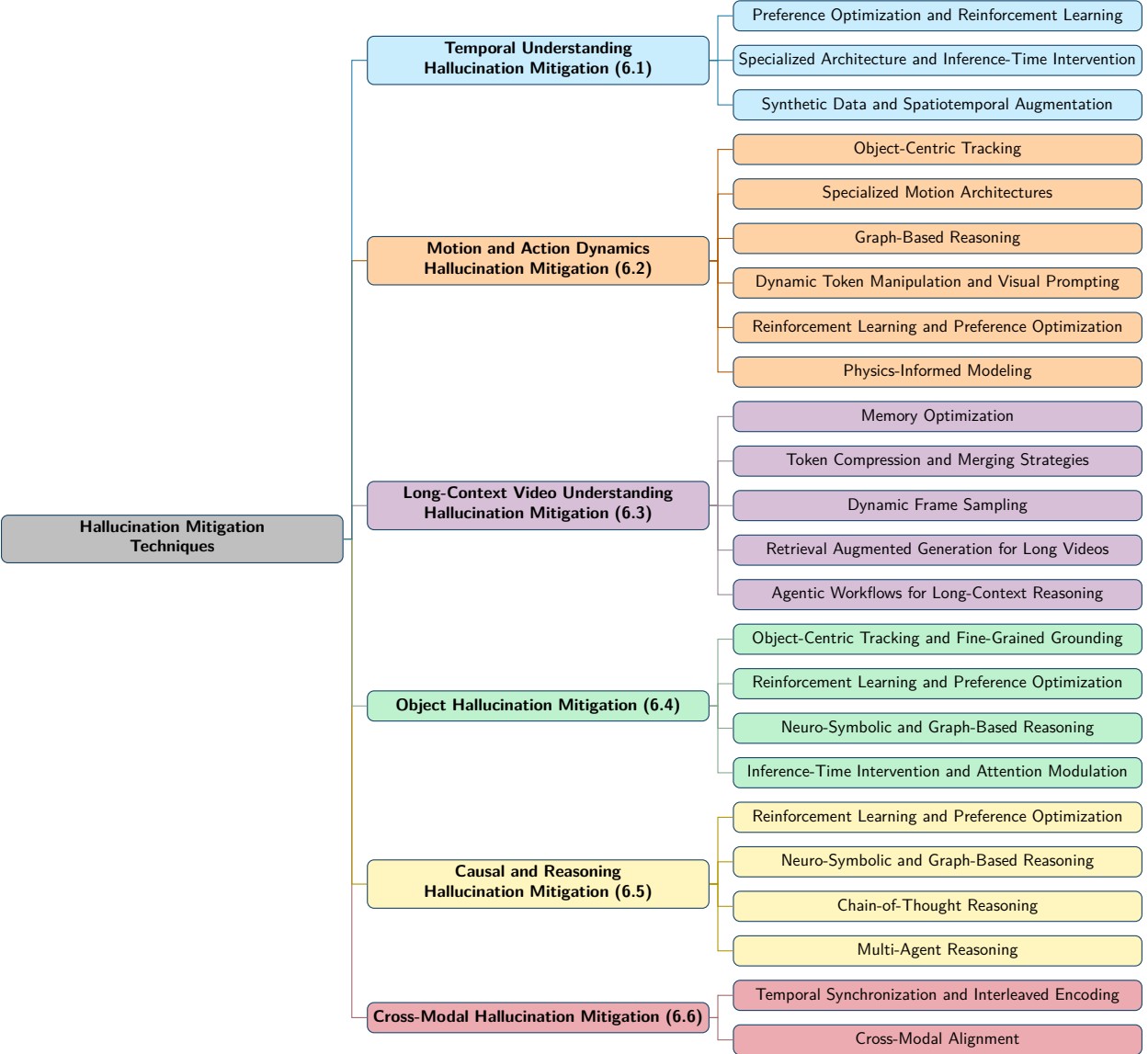

Figure 5: Taxonomy of video understanding hallucination mitigation techniques, organized by category and method.

Time-R1 (Wang et al., 2025j) post-trains Vid-LLMs using RL with verifiable rewards (RLVR), combining a timestamp-aware Intersection-over-Union (tIoU) signal with a reasoning-template reward optimized via Group Relative Policy Optimization (GRPO). On Charades-STA (Gao et al., 2017), Time-R1 achieves R1@0.7 of 35.3, roughly doubling TRACE and nearly tripling TimeChat. Temporal grounding improves when the model is rewarded for evidence-consistent temporal decisions rather than matching surface-form annotations. TimeZero (Wang et al., 2025k) demonstrates that structured RL-guided reasoning can yield competitive temporal localization from as few as 2,500 training samples. However, both methods incur higher training and inference costs and struggle with ultra-long video contexts.

D$^2$VLM (Zeng et al., 2025b) addresses the entanglement between temporal grounding and text generation by decoupling these objectives through explicit evidence tokens that capture event-level visual semantics, forcing the model to obtain temporal evidence before generating an answer. This improves grounding accuracy over TimeChat-7B (Ren et al., 2024a) with only 1.4% computational overhead per token, though performance on episodic memory and fine-grained event boundary detection remains limited.

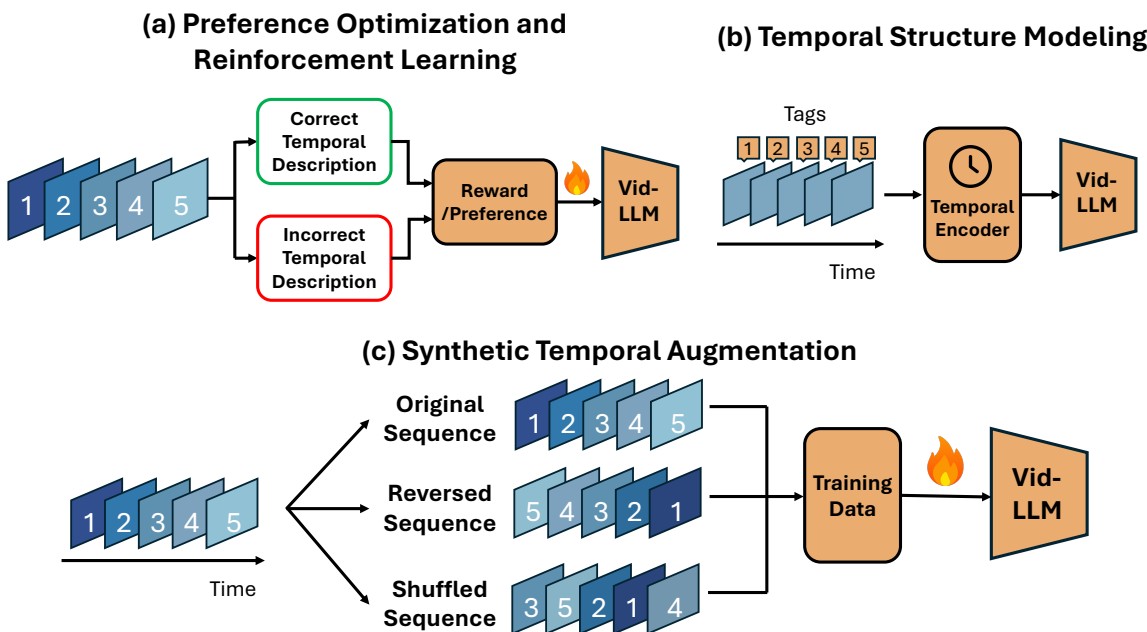

Figure 6: Illustration of representative strategies for mitigating temporal understanding hallucinations in Vid-LLMs. (a) Preference optimization and reinforcement learning methods enforce temporally grounded reasoning by rewarding correct frame ordering and penalizing temporally inconsistent predictions. (b) Temporal structure modeling introduces dedicated temporal encoders or time-aware representations that explicitly capture event ordering and timestamp information for downstream reasoning. (c) Synthetic temporal augmentation improves robustness by exposing the model to temporally perturbed training samples, such as reversed or shuffled event sequences.

Subsequent work refines the RL approach along several axes. VideoChat-R1 (Li et al., 2025g) extends to multi-task reinforcement fine-tuning, jointly optimizing temporal grounding, object tracking, and question-answering, yielding gains over Video-R1 (Feng et al., 2025). MOSSChat-V (Tao et al., 2025) addresses "temporal hacking", where models bypass temporal reasoning to guess outcomes directly, through a process-reasoning reward based on subsequence Dynamic Time Warping (DTW) that supervises intermediate temporal logic. Li et al. (2025c) combine discrete semantic rewards with continuous tIoU signals alongside variance-aware data selection, surpassing Video-R1 on temporal grounding with substantially less training data.

Additional efforts explore hybrid off-policy sampling and soft advantage estimation (Li et al., 2025i), multi-task RL with task-specific localization rewards (Wu et al., 2025c), and metric-based direct preference optimization through chain-of-tasks decomposition (Lee et al., 2025). These studies confirm a shift from static supervised objectives toward adaptive, reward-driven post-training. The main strength is its flexibility: post-training alignment can layer on top of existing Vid-LLMs without architectural redesign. On the other hand, reliable rewards are very tricky to design, long-horizon reasoning increases training instability, and scalability to ultra-long contexts remains very challenging. Because temporal correctness can be turned into verifiable rewards such as tIoU, this is among the strongest levers against temporal hallucination, directly optimizing the chronological decisions that token-level losses leave unconstrained.

**Specialized Architecture and Inference-Time Intervention**   Architectural approaches modify what the model *represents* by embedding temporal structure directly into its token vocabulary, attention mechanisms, or decoding pipeline. Their thought was, if time is only weakly encoded in the input, no amount of post hoc alignment can fully recover precise chronology.

Early work VTimeLLM (Huang et al., 2024a) introduced a three-stage curriculum that progressively sharpens temporal perception through dedicated boundary detection tasks, improving temporal grounding over Video-LLaMA (Zhang et al., 2023b). Models that recognize content well may still fail to segment event boundaries, indicating that temporal hallucination is partly representational. However, VTimeLLM represents timestamps as plain text and relies on detected shot boundaries, limiting intra-shot precision.

LITA (Huang et al., 2024b) combines relative time tokens with a SlowFast architecture, compressing the token count from 25,600 to 356 while nearly doubling mean IoU (mIoU) on ActivityNet-RTL (Huang et al., 2024b). Chrono (Rodriguez et al., 2024) shows that a simpler, model-agnostic approach—interleaving frame embeddings with absolute integer timestamps—can outperform complex specialized modules with only 19M trainable parameters via Low-Rank Adaptation (LoRA). These results suggest that some temporal hallucinations arise not from weak visual features but from the model's inability to interpret temporal metadata. Grounded-VideoLLM (Wang et al., 2024a) extends this with a dual-stream architecture that encodes spatial and temporal components through discrete temporal tokens sharing an embedding space with the language model.

VTG-LLM (Guo et al., 2025c;c) tackles concept shifts from identical output targets for varying inputs by integrating sequence-time embeddings, absolute-time tokens, and slot-based compression enabling denser frame sampling without expanding the context window. GeLM (Chen et al., 2025c) augments the vocabulary with dynamic grounding token pairs fused with visual features through a dual-branch saliency-similarity module, achieving state-of-the-art on ActivityNet-RTL (Huang et al., 2024b) relative to TimeChat (Ren et al., 2024a), which is important for settings where the answer depends on several disjoint moments.

Two recent directions go beyond discrete tokens. VTune (Jung et al., 2025) reformulates temporal grounding as a verification process requiring models to confirm or correct query-moment alignments at inference time, substantially improving TimeChat's consistency on Charades-CON (Jung et al., 2025)—revealing that specialized timestamp encoding alone does not guarantee robust temporal comprehension. DisTime (Zeng et al., 2025c) models time as continuous probability distributions rather than discrete tokens: its distribution-based decoder handles boundary ambiguity while adding less than 1% additional parameters, surpassing discrete-quantization methods on Charades-STA (Gao et al., 2017). Continuous decoding is better suited to uncertain or gradually evolving events, though it remains constrained by frame-level evidence accuracy.

Complementary refinements include iterative timestamp refinement with auxiliary regression losses (Wang et al., 2024h). The field traces a trajectory from text-based timestamps through discrete tokens and dual-stream architectures to continuous distributional modeling. Architectural methods offer stronger temporal inductive bias than post-training alignment, but require more invasive modifications or denser visual processing. A promising direction is unifying these encoding paradigms for fine-grained boundary localization, multi-hop reasoning, and verification-based consistency without task-specific modifications. Embedding time into the representation is a decisive contributor here, since chronology that is never encoded cannot be recovered downstream; by contrast, the inference-time verification variants play only a limited role, which is far smaller than inference-time reasoning methods do for causal hallucination, because they can confirm or correct timestamps the architecture already exposes but cannot, on their own, inject temporal structure.

**Synthetic Data and Spatiotemporal Augmentation**   Data-centric approaches tackle temporal hallucinations at their source: the training distribution. These methods create supervision that foregrounds order, progression, and temporal coherence through synthetic or augmented videos with controlled temporal variations.

Sparrow (Yin et al., 2024) augments real videos with synthetic text-to-image sequences—rendering instructions as image sequences mimicking video frames—reducing required training samples by 85% while improving temporal reasoning on TempCompass (Liu et al., 2024e) over Video-ChatGPT (Maaz et al., 2023). VISTA (Ren et al., 2024b) and COSA (Chen et al., 2023b) address models trained on short clips that fail on long videos by constructing pseudo long-form sequences from short samples, strengthening event-level alignment without large native long-video corpora.

TimeWarp (Vani et al., 2025) generates preference pairs by shuffling and reversing event sequences within video narratives, extending the STIC (Deng et al., 2024) self-training framework to video. This reduces

**(a) Object-Centric Tracking**   **(b) Specialized Motion Architectures**   **(c) Graph-Based Reasoning**

**(d) Dynamic Token Manipulation and Visual Prompting**   **(e) Reinforcement Learning and Preference Optimization**   **(f) Physics-Informed Modeling**

Figure 7: Illustration of representative methods for mitigating motion and action dynamics hallucinations in Vid-LLMs. Each panel depicts a representative strategy corresponding to one category discussed in this subsection, including object-centric tracking, specialized motion architectures, graph-based reasoning, dynamic token manipulation and visual prompting, reinforcement learning and preference optimization, and physics-informed modeling.

hallucination rates by 7.5% on TempCompass (Liu et al., 2024e) at a cost of approximately 60 A100 GPU hours per model. It is particularly effective against shortcut learning, where a model answers correctly from scene cues while ignoring chronology. Complementary works inject temporally sensitive tasks to reduce reliance on static visual bias (Wang et al., 2025o; Zhou et al., 2025a). Perturbation-based augmentation exposes whether the model truly understands sequence order, though aggressive perturbations can drift from realistic video dynamics.

A third branch shifts from recognition to prediction. Next-Event Prediction (NEP) (Wang et al., 2025b) trains models to generate textual summaries of future video segments conditioned on past frames, forcing internalization of event dynamics and causal progression. DIBS (Wu et al., 2024a) pursues a similar goal through diverse LLM-generated event-centric pseudo captions with online boundary refinement.

T3 (Li et al., 2024b) finds that the temporal reasoning bottleneck often resides in the language model rather than the visual encoder: by synthesizing diverse temporal tasks in pure text from image-text datasets, T3 surpasses ShareGPT4Video (Chen et al., 2024c) on TempCompass (Liu et al., 2024e) without any video data.

Data-centric approaches are more scalable than RL and less invasive than architectural redesign, but if synthetic supervision contains temporal bias or implausible event chains, it can regularize the model in the wrong direction. Careful control over temporal structure—through perturbation, composition, or modality transfer—proves more important than sheer data volume. Perturbed-order supervision reliably exposes shortcut learning, making this a solid contributor; it stops short of being decisive only because its benefit is bounded by how faithfully synthetic sequences mimic real temporal dynamics—a ceiling that keeps data augmentation a complement to, not a substitute for, reward- and architecture-based methods.

## 6.2 Motion and Action Dynamics Hallucination Mitigation

Motion and action dynamics hallucination emerges when Vid-LLMs fail to capture fine-grained temporal evolution, object trajectories, and physically grounded motion patterns. We group existing mitigation tech-

niques into six categories (see Figure 7): (i) *object-centric tracking*; (ii) *specialized motion architectures*; (iii) *graph-based reasoning*; (iv) *dynamic token manipulation and visual prompting*; (v) *reinforcement learning and preference optimization*; and (vi) *physics-informed modeling and 4D reconstruction.*

**Object-Centric Tracking**   Object-centric tracking replaces coarse frame-level features with explicit per-entity trajectory representations, since action errors often stem from the model not knowing who moved, where, and in what order. PiTe (Liu et al., 2024d) tracks three key points per object across frames and projects these trajectories into the language embedding space via a dedicated trajectory projector. Supported by an automatic annotation pipeline yielding the PiTe-143k dataset and $k$-means++ clustering to manage computational overhead, PiTe demonstrates superior zero-shot performance on video question answering, temporal grounding, and dense captioning compared to static-alignment baselines such as Video-LLaMA (Zhang et al., 2023b). However, its reliance on sparse keypoint tracking limits performance with small or heavily occluded objects.

VideoOrion (Feng et al., 2024) extends to full object-centric tokenization through a two-branch design: a video-centric branch captures global context while an object-centric branch leverages a detect-segment-track pipeline (GroundingDINO, SAM, XMem) to extract disentangled object tokens. This substantially improves performance across MVBench (Li et al., 2024a), Perception-Test (Patraucean et al., 2023), and ActivityNet-QA (Yu et al., 2019) over spatial-temporal pooling baselines, at a roughly 38% increase in computation time.

iMOVE (Li et al., 2025e) introduces the first motion-aware instruction-tuning dataset (iMOVE-IT) with fine-grained instance spatiotemporal annotations, paired with Event-aware Spatiotemporal Efficient Modeling and Relative Spatiotemporal Position Tokens. This reduces visual tokens by over 34% via adaptive pooling while preserving motion fidelity, and unifies spatial grounding, temporal grounding, and dynamic captioning through mutual-supervision tasks. However, fixed event segmentation granularity limits applicability to ultra-long videos. The key lesson: object grounding is most effective when architecture and training data are jointly designed around motion.

Tracklet-phrase contrastive alignment (Chang et al., 2025) improves captioning fidelity over Vista-LLaMA (Ma et al., 2023). Complementary works include trajectory-guided attention for detecting physical implausibility (Motamed et al., 2025) and slot-based tokenization of object dynamics (Xu et al., 2024a). MotionBench (Hong et al., 2025) exposes remaining limitations on fine-grained motion tasks. These approaches force temporal reasoning to be indexed by entities instead of latent global tokens; their main limitation is scalability in crowded scenes, under heavy occlusion, or with amorphous motion regions. Indexing reasoning to explicit per-entity trajectories makes this one of the strongest remedies, since motion hallucination arises precisely when the model loses track of who moves where and when.

**Specialized Motion Architectures**   Whereas object-centric tracking augments the input representation, specialized motion architectures modify the model's internal processing pathways to better capture high-frequency inter-frame dynamics. Rasekh et al. (2025) observed that joint spatiotemporal attention, as in InternVideo2-Chat (Wang et al., 2024i), is insufficient for distinguishing temporally mirrored actions. Their STAVEQ2 integrates stacked temporal attention (STA) modules within the vision encoder, processing patch embeddings across frames before tokens reach the language model, achieving up to 5.5% accuracy gain on MVBench (Li et al., 2024a) with up to four times fewer attention heads. Early temporal processing is critical when mirrored or near-identical actions must be distinguished, though performance degrades on extremely long videos.

MASH-VLM (Bae et al., 2025) instead targets entangled causal attention within the language model: DST-attention restricts direct interactions between spatial and temporal tokens, while Harmonic-RoPE harmonizes positional IDs across multimodal tokens. This reduces hallucination rates by 12–17% on UNSCENE (Bae et al., 2025) but remains constrained by fixed token budgets beyond 100 frames.

Complementary refinements include: a GOP encoder jointly fusing RGB and motion features within group-of-pictures units (Zhao et al., 2025c), surpassing Video-LaVIT's (Jin et al., 2024) concatenation-based fusion; a diffusion-powered continuous video tokenizer preserving spatial-temporal dynamics (Ge et al., 2024); an

appearance-free optical flow stream with action-centric contrastive learning (Chen et al., 2023a), improving action recognition accuracy by over 3%; and disentangled frame-level and motion-level features through noun-verb grounding (Liu et al., 2024b).

At the systems level, Li et al. (2025h) enable 16 FPS processing via a high-frame-rate aligner with block matrix decomposition, while Hu et al. (2025b) improve fine-grained sensitivity through self-supervised fragment-level tasks. The central trade-off is computational: motion-specialized encoders improve fidelity but increase memory use, reduce compatibility with generic backbones, or degrade on very long videos. Processing high-frequency inter-frame dynamics inside the encoder attacks the root cause directly, a primary contributor for distinguishing visually similar actions.

**Graph-Based Reasoning** Graph-based methods construct explicit structured representations, such as scene graphs where nodes encode objects and edges capture spatial or temporal relationships, to enable reasoning about fine-grained interactions rather than implicit holistic feature matching. Finsta (Fei et al., 2024b) represents both texts and videos as fine-grained scene graph structures unified into a holistic graph, with a spatial-temporal Gaussian differential Graph Transformer that differentiates between moving and stationary objects (Sun et al., 2022). This improves long-form video QA accuracy by 7–9% over the base Vid-LLM, but its performance degrades with noisy annotations and videos containing more than eight sequential actions.

GHR-VQA (Brilli et al., 2025) constructs human-rooted video-level graphs linking frame-level scene graphs via human nodes to a global root, with a hierarchical Conditional Relational Network and HetEdgeGAT encoder for cross-frame message passing. This achieves substantial gains on AGQA (Grunde-McLaughlin et al., 2021) over prior graph-based approaches (Khan et al., 2023), since many motion questions are implicitly actor-centric. However, the single-root constraint limits scalability to multi-actor scenes.

Video-STR (Wang et al., 2025h) bridges graph-based reasoning with RL by constructing inter-object relation graphs and training via GRPO with task-specific rewards. Its rotation-invariant graph representation addresses the lack of explicit spatial rewards in Video-R1 (Feng et al., 2025), though graph construction introduces computational overhead.

Further extensions in this category include: a dynamic graph transformer modeling object-level temporal dynamics (Xiao et al., 2022) beyond the static graphs of HQGA (Peng et al., 2022); temporally coherent event graphs preserving trajectory and velocity attributes (Masala & Leordeanu, 2025); multi-scale temporal modeling with visual-action semantic-aware modules (Liu et al., 2025a); relation-aware graph attention mechanisms (Wang et al., 2023c); and motion-focused video-language learning with verb-variation strategies (Doughty et al., 2024). Graph-based approaches excel at compositional questions involving contact, order, or multi-object dependencies, but noisy detections and annotation errors can themselves become a source of hallucination. Hence, graph reasoning is most useful when paired with stronger perceptual front ends.

**Dynamic Token Manipulation and Visual Prompting** A complementary line of work manipulates visual token representations or augments input with explicit visual prompts that direct attention toward active motion regions. DynImg (Bao et al., 2025) composes key frames into a single dynamic image while using non-key frames as pixel-level temporal prompts highlighting rapidly changing regions, with a 4D rotary positional embedding preserving spatiotemporal order. Unlike Video-ChatGPT (Maaz et al., 2023) and Video-LLaMA (Zhang et al., 2023b), DynImg enables fine-grained pixel-level interactions before feature extraction, achieving substantial gains on MVBench motion subtasks. It performance starts to degrade when more than four non-key frames are used.

MotionSight (Du et al., 2025) decouples object motion from camera motion at zero-shot inference time, since models often hallucinate object movement when the real cause is so-called ego-motion, i.e., the motion of a camera relative to its environment. This yields a 14.3% gain on camera motion tasks in MotionBench (Hong et al., 2025) with modest additional inference latency. ST-LLM (Liu et al., 2024c) shows that LLMs can directly model spatial-temporal token sequences without additional temporal modules, using dynamic masking for masked video modeling that improves MVBench (Li et al., 2024a) accuracy over module-heavy

approaches (Luo et al., 2023). Domain-specific heuristic prompts (Yu et al., 2024b) offer lightweight alternatives. These methods are training-free with minimally invasive front-end corrections. On the other hand, when motion salience is ambiguous, manually designed prompting can suppress useful context as easily as it highlights relevant change. These training-free interventions help moderately, but contribute less than inference-time methods do for causal reasoning: there, chain-of-thought and multi-agent verification restructure the failing inference itself, whereas here prompting can only redirect attention toward motion regions without adding the inter-frame dynamics the model fails to perceive.

**Reinforcement Learning and Preference Optimization**  RL and preference optimization penalize incorrect action dynamics through motion-sensitive reward signals. Standard final-answer rewards are too coarse to penalize subtle mistakes in trajectory or event ordering. Invert4TVG (Chen et al., 2025l) augments RL training with verb completion, action recognition, and video description, which are derived from grounding annotations without external supervision. It reinforces action-semantic consistency while reducing GPU memory by 15%.

VideoCap-R1 (Meng et al., 2025) extends RL to open-ended captioning through GRPO with a structured thinking paradigm, using dual reward mechanisms that includes an LLM-free think scorer and an LLM-assisted caption scorer, to yield strong F1 improvements on CAREBENCH (Xu et al., 2024b) with only 1.5k training samples. Video-CoM (Rasheed et al., 2025) introduces a Chain of Manipulations enabling iterative visual interactions through temporal segment finding, keyframe isolation, and spatial zooming, supervised by step-level rewards for temporal IoU, frame recall, and spatial IoU. It is effective because motion understanding is often a sequential search problem.

VideoPerceiver (Zhao et al., 2025a) enhances temporal sensitivity through key-information-absent video construction and contrastive learning, then applies a comparative GRPO reward enforcing superior response quality from complete videos over temporally degraded inputs. This yields a 22.9% improvement on VRU-Accident (Kim et al., 2025) with considerable computational overhead. VDCAgent (Wang et al., 2025e) eliminates dependence on expensive external models via agentic self-reflection and curriculum DPO, while OwlCap (Zhong et al., 2025) proposes a Caption Set Equivalence Reward jointly optimizing correctness and completeness beyond motion-only (Wang et al., 2024c) or event-coverage-only rewards.

These works together represent a shift from task-specific binary rewards to rewards that capture subtle physical plausibility constraints (trajectory consistency, velocity coherence, biomechanical feasibility, etc.) together with external annotators at affordable costs. Motion-sensitive rewards add a useful but secondary contribution: unlike temporal grounding or causal reasoning, where rewards such as tIoU or evidence-consistency scores directly verify the failing decision, motion rewards only penalize wrong trajectories after the fact and cannot supply the fine-grained dynamics representation the model lacks, making RL a weaker lever here than for those categories.

**Physics-Informed Modeling**  Physics-informed modeling directly embeds geometric and physical priors into the reasoning pipeline, constraining the hypothesis space so that implausible motion interpretations are rejected. Zhou et al. (2025c) address Vid-LLMs' inability to perform dynamic spatial reasoning by introducing the DSR Suite framework with an automated training data pipeline and a Geometry Selection Module employing two stacked Q-Formers. The module condenses question semantics and selectively extracts task-relevant geometric knowledge from 4D reconstruction priors, introducing only 32 geometry tokens to balance information density and computational cost. MASS (Wu et al., 2025d) extends this with depth-based 3D encoding, visual grounding, and an explicit motion tracker. PhyVLLM (Zhan et al., 2025) further extends them with continuous-time modeling through motion-appearance disentanglement, Neural ODE-based dynamics, and self-supervised physical consistency losses.

The difficulty lies upstream: physics-informed methods depend on the quality of depth estimation, reconstruction, or motion decomposition. When these priors are noisy, the model will inherit more errors than flexible learned features would. The challenge is making integration uncertainty-aware and robust to real-world sensing noise. Geometric and physical priors strongly constrain implausible motion, though the contribution is gated by upstream depth and reconstruction quality.

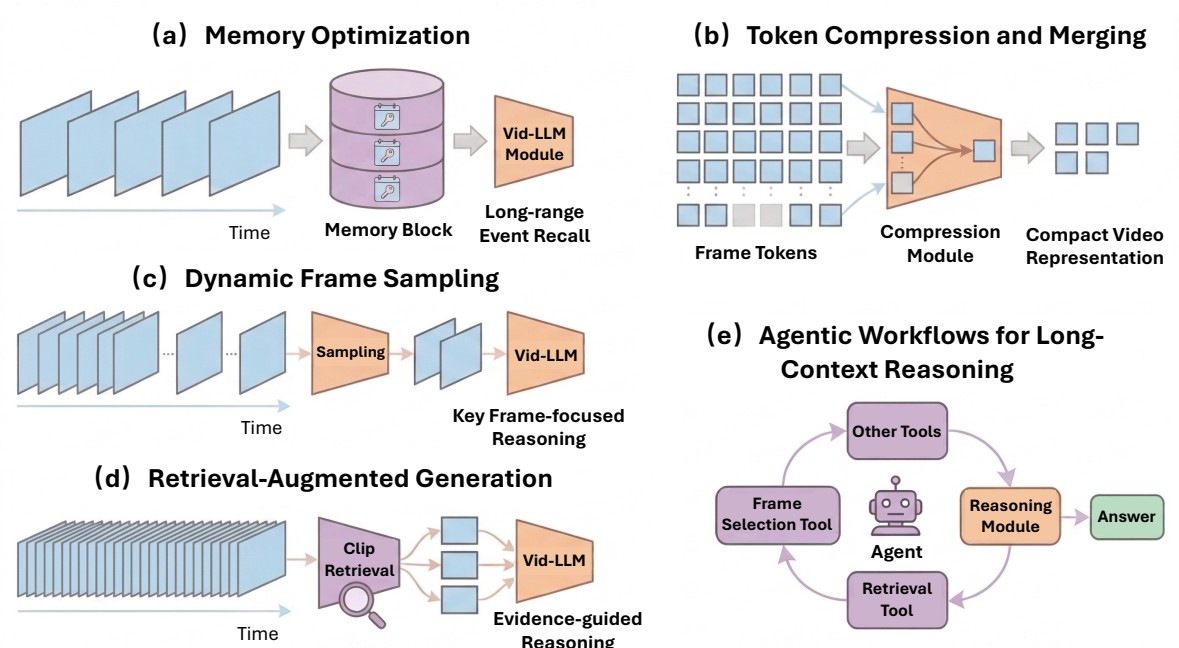

Figure 8: Illustration of representative methods for mitigating long-context video understanding hallucinations in Vid-LLMs. Each panel depicts a representative strategy corresponding to one category discussed in this subsection, including memory optimization, token compression and merging, dynamic frame sampling, retrieval-augmented generation, and agentic workflows for long-context reasoning.

**Summary.** These reviewed mitigations differ in where they intervene: object-centric tracking and specialized architectures strengthen perception; graph-based reasoning and physics-informed modeling impose structure on interpretation; dynamic token manipulation offers cheaper inference-time correction; and RL reshapes training objectives toward motion-sensitive fidelity. Tracking and graph pipelines improve grounding but add preprocessing cost; specialized architectures improve accuracy but reduce efficiency; prompting is lightweight but heuristic; and RL is certainly powerful but expensive. A promising direction is, perhaps, a hybrid design: tracked or prompted motion cues feeding structured reasoning, with lightweight preference signals and uncertainty-aware physical priors.

## 6.3 Long-Context Video Understanding Hallucination Mitigation

As we've mentioned before, long-context video understanding poses a fundamental challenge because exhaustive frame processing is computationally infeasible, so models rely on aggressive sampling, compression, or memory mechanisms. Hallucinations arise from failures to retrieve, retain, or attend to temporally relevant evidence. We break corresponding mitigation techniques into five subcategories: (i) *memory optimization*; (ii) *token compression and merging*; (iii) *dynamic frame sampling*; (iv) *retrieval augmented generation*; and (v) *agentic workflows for long-context reasoning*.

**Memory Optimization**  Early work by Song et al. (2023) introduced a training-free framework combining a memory buffer with merge-based consolidation, but non-learnable frame merging caused over-smoothing in long videos. VideoLLaMB (Wang et al., 2024m) addresses this with recurrent memory bridges, which are learnable memory tokens within bridge layers, alongside a SceneTiling algorithm for semantic video segmentation, ensuring memories respect scene structure rather than arbitrary chunk boundaries. On NIAVH, VideoLLaMB substantially outperformed MA-LMM (He et al., 2024). A concurrent variant (Wang et al., 2024k) refined this for streaming with linearly scaling recurrent memory caches.

Subsequent work shifted toward human-inspired and online memory mechanisms. Flash-VStream (Zhang et al., 2024a) introduced a Spatial-Temporal-Abstract-Retrieved (STAR) memory that dynamically com-

presses visual information online, achieving real-time inference with lower VRAM. An enhanced version (Zhang et al., 2025d) decomposed memory into Context Synopsis Memory for temporal aggregation and Detail Augmentation Memory for spatial detail, enabling sub-second responses. It effectively addresses hallucination that arise when coarse narrative state and local visual detail are compressed. $\infty$-Video (Santos et al., 2025) augments video Q-formers with continuous-time long-term memory consolidation based on a Gibbs density function, dynamically allocating higher memory granularity to relevant segments, though fine-grained temporal reasoning remains limited when sampling is coarse.

Video-SALMONN S (Sun et al., 2025c) replaces token merging with a test-time training layer using Hessian-free optimization to continually update memory representations, processing over one million tokens with fixed memory overhead at approximately 14% additional computational cost. VideoMem (Jin et al., 2025c) addresses the rigidity of static memory with adaptive memory management trained via Progressive Grouped Relative Policy Optimization (PRPO) with Temporal Cascading Rewards, achieving a $3.1\times$ training speedup. The key insight is: long-context faithfulness depends less on stored tokens than on learning when to retain, rewrite, or discard them.

A parallel line of work targets KV cache compression for streaming. Yang et al. (2025c) introduced query-agnostic KV cache compression with fixed memory footprint, while Chen et al. (2025g) replaced uniform segmentation with dynamic semantic partitioning. Xu et al. (2025a) aligned training with streaming inference via overlapped short-chunk attention, achieving competitive performance against GPT-4o mini on infinite-stream evaluation. Chen et al. (2025i) developed MR-SP to scale RL training to hour-long videos on a single A100 node via sequence parallelism with cached video embeddings.

Additional strategies include multi-stage long-context training curricula (Xue et al., 2024), hybrid Mamba-Transformer architectures for linear-complexity processing (Ren et al., 2025), task-aware KV sparsification (Qin et al., 2025b;a), and agentic frameworks with adaptive token selection (Yang et al., 2025b). The central trade-off remains between memory footprint and retrieval fidelity: aggressive compression risks losing transient events critical for fine-grained reasoning, while richer memory representations undermine real-time applicability. Because long-context hallucination is fundamentally a retention problem, learning what to keep, rewrite, or discard is a primary contributor.

**Token Compression and Merging Strategies**   Token compression reduces representation size by fusing or aggregating visual tokens after encoding. The difficulty is that naive pooling often destroys temporal order, object identity, and scene transitions, which really matter for faithful reasoning.

LongVLM (Weng et al., 2024) decomposes long videos into segments and applies hierarchical token merging, reducing token counts by roughly two orders of magnitude, but its fixed compression ratio limits adaptability. Dynamic-VLM (Wang et al., 2024b) adaptively allocates between 16 and 576 tokens per frame based on video length, substantially improving over LLaVA-OneVision but remaining constrained by a fixed 12K token budget. ReTaKe (Wang et al., 2024f) jointly reduces temporal and knowledge redundancies through DPSelect for keyframe selection and PivotKV for KV cache compression. It achieves 3–5% accuracy gains on Video-MME (Fu et al., 2025) over comparable models, though degrading on fine-grained tasks such as Needle QA.

Video-XL (Shu et al., 2025) leverages LLM attention sparsity through Visual Summarization Tokens that adjust compression granularity based on information density. Wei & Chen (2024) adapt YaRN position embedding extension to visual tokens with progressive pooling, reducing memory by approximately 45% without retraining. BIMBA (Islam et al., 2025) is the first state-space model (SSM)-based token compression approach, which uses bidirectional selective-scan at $O(N)$ complexity and compressing tokens by 16 times. CrossLMM (Yan et al., 2025) decouples visual token processing from the LLM through dual cross-attention, producing text-conditioned compressed representations with sub-linear memory growth. Dahal et al. (2025) combine temporal pooling with DPO, compressing video into a single visual token per second with a 23 times token reduction, though reliability depends on subtitle quality.

Liu et al. (2025e) introduced reconstructive token compression extending processing to over 8,000 frames while maintaining near-perfect Needle-in-a-Haystack accuracy. This work demonstrates that reconstructive compression preserves semantics better than pure sparsification. Jiang et al. (2025a) insert a Mamba-based

temporal encoder between image encoder and LLM, showing compression can be built into the sequence processor itself. Complementary strategies include hierarchical caching for streaming (Wang et al., 2025m), thumbnail-and-sampling schemes (Qu et al., 2024), and strictly causal two-stage compression (Chen et al., 2025f). Token compression is better suited than dynamic sampling when broad temporal coverage must be preserved, but risks semantic averaging where distinct events collapse into a single token. Fitting more frames into a fixed budget while preserving event structure directly attacks the capacity bottleneck, a core contributor for long videos.

**Dynamic Frame Sampling**   Dynamic frame sampling reduces information overload at the input stage by selecting or filtering frames based on semantic relevance or information density, thus preventing redundancy-induced fabrication before tokens enter the model.

The LIVE framework (Chen et al., 2024a) introduced Streaming EOS prediction for streaming dialogue, dynamically determining when to respond or remain silent. However, its single CLS token per frame limits spatial understanding. VideoLLM-MoD (Wu et al., 2024b) introduces a Mixture-of-Depths strategy that selects the top-$k$ most informative vision tokens per frame while routing others through residual connections, reducing FLOPs by 42% and supporting 1.7× longer context windows. Sun et al. (2024) combine CLIP-score-guided frame sampling with question-injected visual features, achieving a hallucination rate reduced by over 24% relative to MovieChat (Song et al., 2023). It comes from the key insight that relevance should be defined with respect to the question, not only video salience. SEAL (Wang et al., 2024d) decomposes long videos into compact scene, object, and action tokens with an Attention Learning module maximizing query relevance and token diversity.

Sun et al. (2025b) extended keyframe extraction to temporally coherent key clips with adaptive resolution, outperforming frame-level baselines on Video-MME and MLVU (Zhou et al., 2025b). Yao et al. (2025) reformulated keyframe selection as clip-to-frame prediction preserving scene continuity. Xu et al. (2025b) leveraged event-camera density for coarse-to-fine redundancy removal at lower cost. Chasmai et al. (2025) introduced moment sampling through a pre-trained moment retrieval model, yielding consistent gains across four long-form VideoQA benchmarks. Ok & Lee (2025) showed that vision encoder confidence scores do not correlate with downstream performance and that up to 50% of frames can be removed without significant degradation, motivating segmentation-area-based selection as a more robust alternative.

The design space continues to grow: hierarchical visual retrieval (Gao et al., 2025a), RL-based temporal sampling (Tang et al., 2025a), DPP-driven diversity-aware selection (Wang et al., 2025f), adaptive frame importance prediction (Li et al., 2025b), differentiable selection policies (Buch et al., 2025), iterative frame spotlighting (He et al., 2025b), and training-free query-aware sampling (Liu et al., 2025d). Dynamic sampling is most effective when redundancy is high and target evidence is localized; it is less effective when essential information is temporally dispersed or only identifiable after deeper reasoning. Removing redundancy before tokens enter the model is a strong, low-cost contributor, since redundancy-induced dilution is a leading cause here.

**Retrieval-Augmented Generation for Long Videos**   RAG treats the video as a searchable knowledge base, extracting only query-relevant segments. This approach is especially effective when only a small fraction of a long video is relevant.

Naive RAG strategies segment videos mechanically into short clips and retrieve them independently, but they disrupt temporal continuity. Vgent (Shen et al., 2025) addresses this with a graph-based framework modeling clips as nodes interconnected by shared entities, outperforming naive RAG by 8.6% on MLVU (Zhou et al., 2025b). MemVid (Yuan et al., 2025) maintains a key-value cache for holistic video memorization with generative context-aware query expansion operating directly on video content, improving MLVU (Zhou et al., 2025b) accuracy by 8.7% with lower GPU memory than Video-XL (Shu et al., 2025). These methods employ retrievals that apply uniform effort regardless of query complexity.

AdaVideoRAG (Xue et al., 2025a) classifies query intent and dynamically routes questions through hierarchical retrieval paths, from direct inference for simple queries to graph-based retrieval for complex multi-hop reasoning. SceneRAG (Zeng et al., 2025a) segments videos into narrative-consistent scenes using automatic

speech recognition (ASR) transcripts and temporal metadata for dynamic knowledge graph construction, since retrieval quality depends on whether indexed units correspond to meaningful event boundaries. Clearly, reliance on ASR quality limits its effectiveness on videos with no or sparse dialogue.

Ranasinghe et al. (2024) show that explicit entities serve as better indexing anchors than redundant captions, outperforming LLoVi (Zhang et al., 2023a). Goulas et al. (2024) replace LLoVi's joint caption processing with a linear-complexity frame-wise approach. Ma et al. (2024) reformulate long video understanding as document retrieval with multi-stage agent interaction. Luo et al. (2024) propose a training-free framework replacing redundant visual tokens with retrieved auxiliary texts (optical character recognition (OCR), ASR, and object detection), demonstrating that auxiliary text retrieval can be more efficient than additional visual tokens (Fan et al., 2024). Additional efforts decompose long videos into short-term captions for LLM-based reasoning (Ataallah et al., 2024). Retrieval-based methods shift complexity from sequence length to indexing quality, but retrieval errors are often unrecoverable, so recent systems increasingly combine retrieval with memory, graph structure, or adaptive routing. When only a fraction of a long video is relevant, retrieval is among the most effective levers, shifting the burden from sequence length to indexing quality.

**Agentic Workflows for Long-Context Reasoning**   Agentic workflows turn video understanding into a multi-step process of planning, searching, observing, verifying, and revising.

Zhang et al. (2025g) framed long-video understanding as autonomous tool-using search over global, clip-level, and frame-level retrieval. It is a shifte from passive ingestion to active evidence seeking. Wang et al. (2025q) replaced caption-heavy preprocessing with direct pixel-level evidence extraction through a plan-observe-reflect loop, reducing token usage.

Several methods strengthen the agentic loop: Li et al. (2025d) leverage multi-granular clue exploration with verification-enhanced reflection; Dong et al. (2025b) let the model identify reasoning gaps and request targeted visual evidence; VideoLucy (Zuo et al., 2025) adds hierarchical memory and backtracking; VideoARM (Yin et al., 2025) integrates observe-think-act-memorize cycles with hierarchical multimodal memory; and GCAgent (Yeo et al., 2025) constructs schematic and episodic memory before answering to prevent event merging.

The literature also has works employing collaborative or uncertainty-aware reasoning. LVAgent (Chen et al., 2025a) distributes tasks across multiple specialized Vid-LLM agents. VideoAgent2 (Zhi et al., 2025) leverages uncertainty-aware CoT to trigger plan adjustment without heavy external pipelines. TimeSearch (Pan et al., 2025) learns adaptive temporal exploration rather than relying on hand-designed policies, while MR. Video (Pang & Wang, 2025) addresses multi-round exploration bottlenecks through a MapReduce-style design analyzing short clips in parallel.

Agentic workflows are especially useful when evidence is sparse or requires iterative disambiguation, while they also introduce latency, planning instability, and possible error propagation. Many successful agentic systems depend on the same components discussed above: retrieval to narrow search, memory to preserve discoveries, and compression to keep iterative inference tractable. Active multi-step evidence seeking contributes strongly yet remains secondary to the memory, retrieval, and compression machinery it orchestrates; unlike inference-time reasoning for causal hallucination, where the agentic loop is itself the primary fix, here it coordinates other long-context components rather than resolving the capacity bottleneck on its own.

**Summary.** Long-context hallucination mitigation has evolved from static compression into adaptive systems deciding what to store, process, retrieve, or revisit. Memory methods excel when continuity matters but require robust retention policies. Sampling and compression improve efficiency but risk removing decisive evidence. Retrieval-augmented methods scale well but remain affected by retrieval misses. Agentic workflows offer the greatest flexibility and quality at the cost of latency, cost, and complexity.

## 6.4   Object Hallucination Mitigation

Object hallucination arises from failures to consistently aggregate object evidence across time, due to motion, occlusion, and appearance variation, which make object perception more complex than in static images. We break the solutions into four subcategories: (i) *object-centric tracking and fine-grained grounding*; (ii)

Figure 9: Overview of representative strategies for mitigating object hallucinations in Vid-LLMs. The panels illustrate four major solution paradigms discussed in this subsection: (a) object-centric tracking and fine-grained grounding for maintaining consistent object representations; (b) reinforcement learning and preference optimization that penalize unsupported object predictions during training; (c) neuro-symbolic and graph-based reasoning that structures entities and relations for grounded reasoning; and (d) inference-time intervention and attention modulation that dynamically steers the model toward visually grounded evidence.

*reinforcement learning and preference optimization*; (iii) *neuro-symbolic and graph-based reasoning*; and (iv) *inference-time intervention and attention modulation*.

**Object-Centric Tracking and Fine-Grained Grounding** The most direct remedy is to force models to explicitly localize, track, and reason about specific entities rather than relying on holistic scene representations. VideoGLaMM (Munasinghe et al., 2024) introduces a dual vision encoder (spatial and temporal) with a spatiotemporal decoder for precise mask generation, substantially improving interrogative mIoU on VidSTG (Zhang et al., 2020) over PG-Video-LLaVA (Munasinghe et al., 2023). Mask supervision forces the model to maintain correspondence between words and localized visual evidence. On the other hand, producing pixel-level masks is expensive, and its consistency struggles with objects of varying granularity across long sequences.

ObjectMLLM (Tang et al., 2025e) encodes bounding boxes and labels as quantized text tokens, nearly doubling accuracy on CLEVRER-MC (Yi et al., 2019) when augmenting VideoLLaMA 2 (Cheng et al., 2024). RGA3 (Wang et al., 2025a) supports arbitrary visual prompts, including masks, bounding boxes, arrows, points, and scribbles, as both input and output, with a Spatial-Temporal Overlay Module propagating prompts across frames, reducing hallucination rates by approximately 10% over VideoRefer (Yuan et al., 2024). Many hallucinations arise not from failure to detect an object once but from failure to maintain its identity over time.

Gu et al. (2025) introduced text-guided temporal sampling and attribute-aware spatial activation, improving grounding on HCSTVG-v1 (Tang et al., 2021) over TubeDETR (Wasim et al., 2023). Gao et al. (2025c) coupled an MLLM with an open-vocabulary detector through a reference-semantic token, boosting m_vIoU from 8.2 to 36.2 on HC-STVG v1. Kazakos et al. (2025) extended grounded captioning to video with spatio-temporal adapters and a temporal objectness head, yielding state-of-the-art msIoU on VidSTG (Zhang et al., 2020). It demonstrates that object-faithful generation benefits from temporal consistency constraints, not only spatial localization.

Complementary works include region-level token marks (Heo et al., 2025), semantically decoupled slots (Xu et al., 2024a), grid-based local-global transcription (Chowdhury et al., 2025), intention-oriented box adapters (Qiu et al., 2025), open-vocabulary detection for validation (Montes & Lozano, 2025), tokenized object dynamics (Feng et al., 2024), and fused multi-encoder architectures (Chung et al., 2025). The consistent principle is that anchoring outputs to verifiable spatiotemporal evidence suppresses perceptual hallucinations. On the other hand, fine-grained grounding often depends on external detectors, dense supervision, or expensive token budgets, which are costly for long, cluttered videos. Anchoring each claim to localized, tracked evidence is the most direct contributor against object hallucination, suppressing unsupported entities at the source.

**Reinforcement Learning and Preference Optimization**  As discussed, preference optimization reshapes the output distribution by penalizing hallucinated entities, attributes, and spatial relations. VideoSAVi (Kulkarni & Fazli, 2024) pioneered self-aligned Vid-LLM training: a self-critiquing mechanism identifies reasoning errors and generates on-policy preference pairs for DPO without external supervision, yielding gains on TGIF-QA (Jang et al., 2017) and other video QA benchmarks. The model is trained to distinguish visually faithful descriptions from unsupported ones. Its iterative self-play is relatively computationally expensive.

VistaDPO (Huang et al., 2025a) introduced hierarchical spatial-temporal alignment at three levels: instance semantics, temporal events, and fine-grained spatial-object grounding. It applies token-level optimization via sequential KL divergence, outperforming coarse baselines on VideoHallucer (Wang et al., 2024l) and EventHallusion (Zhang et al., 2024b). PaMi-VDPO (Ding et al., 2025) eliminates offline data dependence by dynamically generating rejected samples through video augmentations, achieving competitive performance with only 10k samples. VideoPASTA (Kulkarni & Fazli, 2025) constructs targeted adversarial preference pairs contrasting dense and sparse temporal sampling, demonstrating that careful augmentation can rival methods requiring far larger preference-labeled datasets.

Oracle-RLAIF (Shi et al., 2025) replaces trained scalar reward models with an Oracle ranker providing ordinal feedback via $GRPO_{rank}$. It reports outperforming prior RLAIF approaches (Ahn et al., 2024). ViSS-R1 (Fang et al., 2025) shows that self-supervised pretext-task rewards can complement preference-based objectives without architectural modifications. Additional works include QA-driven reward signals (Liu & Wan, 2024), self-consistency-based preference pair construction (Dang et al., 2025b), and detection-and-tracking pipelines with RL policies to minimize false positive object mentions (Shankar & Surendran, 2025).

Preference-based methods are architecture-agnostic. They shape outputs indirectly: if the underlying representation lacks reliable object evidence, optimization may reduce blatant hallucinations but fail on rare objects, heavy occlusion, or fine-grained spatial distinctions. These methods need to be paired with representations that already expose object structure. Preference signals penalize hallucinated entities, but contribute less than RL does for temporal or reasoning hallucination: there a verifiable reward targets the exact failing decision, whereas for object hallucination RL only reshapes the output distribution and cannot create object evidence that a weak visual representation never encoded.

**Neuro-Symbolic and Graph-Based Reasoning**  Neuro-symbolic and graph-based methods impose structural constraints by converting unstructured video into explicit relational graphs enforcing verifiable reasoning over entities. ReGR (Wang et al., 2023c) introduces a dual-graph architecture: a Spatiotemporal Graph Attention Network captures object dynamics, while a Multi-Relation Graph Attention Network encodes fine-grained interactions, with question-guided filtering to suppress irrelevant features. This improved accuracy on TGIF-QA (Jang et al., 2017) FrameQA but degraded on longer videos requiring complex temporal reasoning.

LGQAVE (Rongali et al., 2024) constructs dynamic spatial object graphs conditioned on masked question embeddings via a Question-aware Dynamic Graph Transformer, leveraging MiniGPT-4 for visual grounding. Accuracy improved by 9.29% on average across six video QA benchmarks including TGIF-QA (Jang et al., 2017), surpassing CoVGT (Xiao et al., 2023), which lacks question-specific conditioning. The shift from static to question-conditioned graphs is important since some object hallucinations are due to selecting the

wrong subset of entities for the question at hand. However, computational overhead (289 GFlops) and reliance on MiniGPT-4 limit scalability beyond 600 frames.

Dai et al. (2025) integrated an explicit scene graph predictor, improving MOMA-QA accuracy from 76.62% to 79.66% over SeViLA (Yu et al., 2023). Complementary efforts encode textual scene graph sequences (Linok et al., 2025), leverage dual and heterogeneous graphs (Liang & Sun, 2025), and inject scene graphs into Vid-LLM prompts. What is central in this line of work is that hallucination is reduced by making object claims pass through a structured reasoning bottleneck. Forcing object claims through a structured relational bottleneck contributes strongly, provided scene parsing is accurate.

**Inference-Time Intervention and Attention Modulation**   Inference-time interventions redirect attention toward task-relevant visual evidence during generation, offering a training-free alternative. SFA (Scan, Focus, and Amplify) (He et al., 2025a) integrates an external Video Text Spotting model with a three-stage pipeline: candidate text region detection, adaptive windowing with multimodal scoring, and region enhancement. SFA achieved up to 45.80% improvement over prior methods, including a 10.85% accuracy gain over Qwen2.5-VL-7B on RoadTextVQA (Tom et al., 2023), surpassing GAT (Zhang et al., 2025h). Instead of trusting learned attention maps, it explicitly re-centers decoding on localized visual-textual content, though effectiveness is bounded by the text spotting model's accuracy.

Jung et al. (2023) augmented visual frame representations with externally retrieved textual descriptions, reducing hallucinated object mentions in dense captioning. Inference-time interventions offer zero training cost, modularity, and compatibility with arbitrary base models, but remain narrowly scoped. Their success hinges on reliably identifying the right evidence.

**Summary.**   Object-centric grounding is very direct and faithful, also realatively expensive. Preference optimization is more modular but cannot compensate for weak visual representations. Graph-based reasoning improves interpretability but relies on accurate scene parsing. Inference-time intervention is training-free and relatively narrow-scoped.

## 6.5   Causal and Reasoning Hallucination Mitigation

Causal and reasoning hallucination mitigation addresses failures where models generate unsupported causal relations, intentions, or abstract conclusions. We subcategorize its mitigation techniques as (i) *reinforcement learning and preference optimization*; (ii) *neuro-symbolic and graph-based reasoning*; (iii) *chain-of-thought reasoning*; and (iv) *multi-agent reasoning frameworks*.

**Reinforcement Learning and Preference Optimization**   RL and preference optimization penalize logically inconsistent or visually ungrounded reasoning traces during post-training. Video-R1 (Feng et al., 2025) established the viability of RLVR for video reasoning but also exposed three challenges: holistic captions erasing temporal structure, training data solvable from textual priors alone, and CoT traces grounded in commonsense rather than visual evidence. Subsequent work has systematically addressed these limitation.

VideoRFT (Wang et al., 2025d) introduces a cross-modal CoT refinement stage where an MLLM revises initial reasoning by re-examining the video, with a semantic-consistency reward enforcing visual-textual alignment, yielding consistent gains over Video-R1 (Feng et al., 2025). Video-R2 (Maaz et al., 2025) adds a Temporal Alignment Reward and consistency gating that link reasoning trace claims to timestamps and the final answer, improving think-answer consistency across eleven benchmarks. Addressing *visual thinking drift*, Luo et al. (2025b) proposed the Visual Evidence Reward where an auxiliary judge evaluates whether intermediate steps reference verifiable visual evidence, achieving up to 9.0% accuracy gains across ten video reasoning benchmarks.

Standard binary rewards cannot distinguish partially correct reasoning from fabricated chains. TW-GRPO (Dang et al., 2025a) enables finer-grained credit assignment through token-level importance weighting based on information entropy with multi-level soft rewards, achieving 18.8% improvement over Video-R1. VerIPO (Li et al., 2025j) employs a Rollout-Aware Verifier within an iterative GRPO-Verifier-DPO loop, making DPO training 7× faster than GRPO. Reflection-V (Jian et al., 2025) tackles visual attention decay

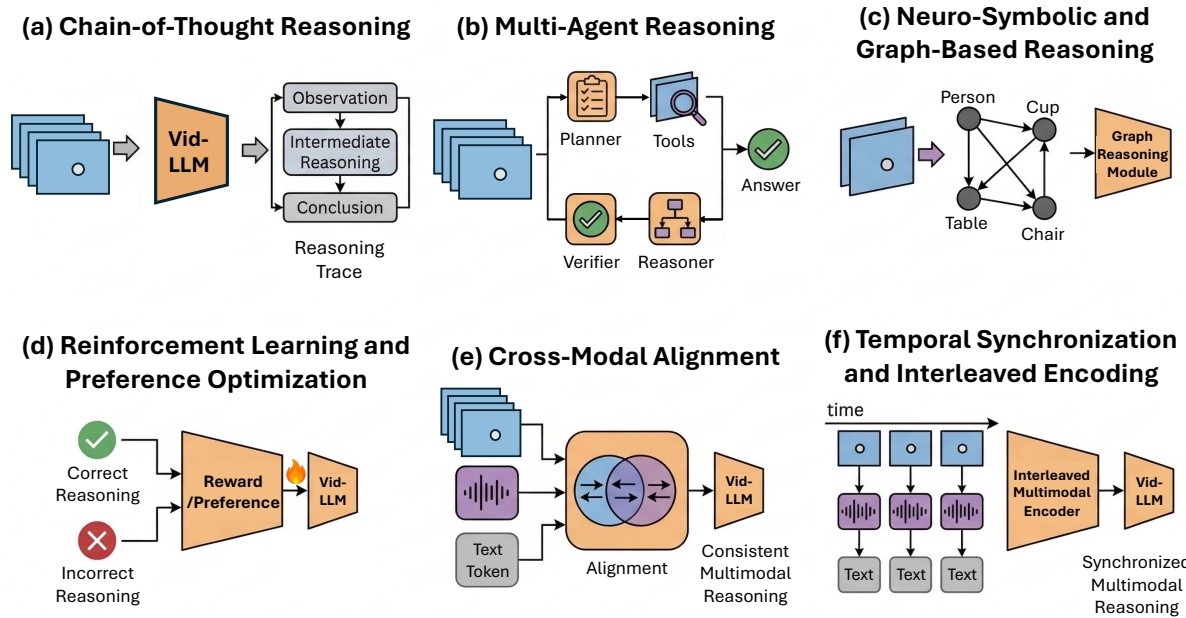

Figure 10: Representative mechanisms for mitigating causal and reasoning, and cross-modal hallucinations in Vid-LLMs. Panels (a)–(d) illustrate reasoning-level mitigation strategies discussed in Section 6.5, including chain-of-thought reasoning, multi-agent reasoning, neuro-symbolic and graph-based reasoning, and reinforcement learning with preference optimization. Panels (e) and (f) present representative approaches for cross-modal hallucination mitigation described in Section 6.6, including cross-modal alignment and temporal synchronization with interleaved multimodal encoding.

in long reasoning chains through cold-start initialization with visual reflection data and a visual attention-based reward.

ReWatch-R1 (Zhang et al., 2025a) pairs multi-agent data synthesis simulating re-watching behavior with an Observation and Reasoning reward, reducing text-only solvability from 68.9% to 29.4%. VidBridge-R1 (Chen et al., 2025e) identifies a conflict between convergent QA and divergent captioning objectives, introducing intermediate proxy tasks (e.g., masked event inference and segment isolation) while eliminating the supervised fine-tuning stage. Conan (Ouyang et al., 2025b) combines multi-scale frame categorization with difficulty-aware progressive training. It reports surpassing Video-R1 by 10%+ across six multi-step reasoning benchmarks.

ReAgent-V (Zhou et al., 2025d) shifts optimization to inference time through a modular multi-agent framework generating real-time rewards via Entropy-Calibrated Relevance Scores, achieving improvements with only 45% of the training data. Complementary efforts include reinforced causal keyframe search (Zhou et al., 2025e), rank-based RLAIF (Shi et al., 2025), and self-training with preference optimization (Zohar et al., 2024). From these works, we can observe progression from simple binary outcome rewards toward sophisticated supervision of intermediate reasoning. Because reasoning hallucination lives in the inference process, rewarding evidence-grounded reasoning traces is among the strongest contributors, directly penalizing fabricated causal chains.

**Neuro-Symbolic and Graph-Based Reasoning**  Neuro-symbolic and graph-based approaches address causal hallucinations by constructing explicit structured representations that ground inference in observable entities and relationships. It makes inferential chains auditable by design.

Some hallucinations are confounding problems where the model mistakes correlation for causality. To alleviate this, EIGV (Li et al., 2022d) partitions visual scenes into causal and environment components through a Structural Causal Model, enforcing equivariant and invariant grounding via contrastive learning, improv-

ing over invariant-only approaches (Li et al., 2022c) on NExT-QA (Xiao et al., 2021). VCSR (Wei et al., 2023) applies causal front-door interventions through a Question-Guided Refiner and Causal Scene Separator, achieving gains on predictive and counterfactual tasks over methods (Liu et al., 2022a) that neglected temporal causal refinement. Lyu et al. (2023) leverage Semantic Role Labeling (SRL) to guide multi-step retrospective-prospective reasoning, dynamically attending to different SRL arguments and selecting relevant frames. This substantially improves TrafficQA (Xu et al., 2021) accuracy over prior graph-convolutional methods, though reliance on SRL parser quality limits generality.

GHR-VQA (Brilli et al., 2025) constructs human-rooted video-level graphs with hierarchical conditional relational networks, efficiently modeling human-object interactions though limited to single-actor scenes. Tang et al. (2024) extend with dynamic graph convolution transformers. Broader explorations include entailment-tree frameworks (Sanders et al., 2024), scene-graph and knowledge-graph fusion (Taluzzi et al., 2025), scene graph-guided self-training (Qiu et al., 2024), theory-of-mind reasoning (Chen et al., 2025k), and causal-aware temporal grounding (Dong et al., 2025a).

The advantages of these methods include transparency and auditable reasoning chains, but they come at the cost of requiring clean scene decomposition, reliable detection and extraction, and accurate parsing. These methods are strongest when the reasoning problem has clear compositional structure and where interpretability is important. Structured representations contribute less to reasoning than to temporal, motion, or object hallucination: in those categories graphs and time-aware tokens encode the very perceptual evidence that is failing, whereas reasoning hallucination occurs in the inference step itself, so structure mainly makes the chain auditable and stays dependent on clean scene decomposition rather than fixing the inference directly.

**Chain-of-Thought Reasoning** CoT reasoning decomposes complex inferential tasks into intermediate steps, grounding each logical transition in observable evidence. Note that free-form textual CoT (which ignores visual evidence) can actually worsen hallucination here. Only when intermediate reasoning is visually grounded does the CoT reduce hallucination.

ViTCoT (Zhang et al., 2025i) uses a video-text interleaved paradigm that re-examines key frames at each reasoning step, achieving 5.5% improvement over baseline CoT. Causal questions often require returning to evidence after a hypothesis is formed. ChainReaction (Parmar et al., 2025) tackles causal-why video QA through a modular two-stage architecture: a Causal Chain Extractor generates chains grounded in Structural Causal Models, and a Causal Chain-Driven Answerer derives responses. While outperforming monolithic baselines, approximately 38% of errors originate from misinterpretation of object roles during chain extraction.

CoT-Vid (Jin et al., 2025a) introduces dynamic inference path routing determining whether a question requires multi-step reasoning or direct answering, with video self-consistency verification. This training-free framework outperforms Video-R1 by up to 9.3%, avoiding unnecessary reasoning cost on simple queries.

Liu et al. (2025b) introduce video-grounded entailment tree reasoning. It achieves a competitive performance with 257× fewer parameters than VideoAgent (Wang et al., 2024g). Liang et al. (2025) refine imperfect MLLM-generated chains via a three-phase filter-and-learn pipeline. Wang et al. (2023a) replace visual embeddings with text descriptions and a token bottleneck, achieving 5× inference speedup. Other works include perception-to-cognition pipelines (Fei et al., 2024a), atomic task planning (Chen et al., 2025j), reasoning chain distillation (Shi et al., 2024), sub-task-oriented instruction tuning (Wang et al., 2025i), counterfactual thought chains (Zhang et al., 2025j), and decomposition-based subquestions (Kamoto et al., 2025). From these works we can observe that CoT helps only when the chain remains tethered to visual evidence.

**Multi-Agent Reasoning** Multi-agent frameworks extend CoT to collaborative, iterative processes where specialized agents debate, verify, and refine intermediate inferences, directly targeting error propagation where an early causal misattribution corrupts all downstream conclusions.

MoReVQA (Min et al., 2024) decomposed video QA into fixed stages that proved brittle when queries did not conform. CAViAR (Menon et al., 2025) shifts to adaptive, iterative execution: a reasoning agent dynamically composes video-processing modules based on intermediate results, and then a natural-language

critic evaluates sampled reasoning traces. CAViAR achieves 77.2% on LVBench (Wang et al., 2025g) versus 48.7% for direct inference.

From different perspectives, Xie et al. (2025) orchestrate diverse prompting strategies with an MLLM-based evaluator for compositional and causal reasoning without task-specific training. ReAgent-V (Zhou et al., 2025d) generates evaluation reports during inference as dynamic rewards, employing reflection across conservative, neutral, and aggressive perspectives. VideoAgent (Fan et al., 2024) frames video reasoning as iterative collaboration where specialized agents collaboratively select, analyze, and refine reasoning through reward-driven feedback.

Once again, multi-agent collaboration offers clear advantages in error recovery and reasoning diversity, but also introduces coordination overhead, latency, and reliance on critic agents. This is an active line of research because mitigating causal hallucination require not just a better model but a better reasoning process. Collaborative verification contributes strongly by catching early causal misattributions before they propagate, at the cost of coordination overhead.

**Summary.** RL reshapes model behavior through reward-driven post-training; neuro-symbolic methods make reasoning auditable; CoT externalizes intermediate logic for verification; and multi-agent systems enable procedual error correction. Clearly, a promising direction is to combine verifier-trained rewards, structured representations, evidence-grounded reasoning traces, and lightweight agentic verification.

### 6.6 Cross-Modal Hallucination Mitigation

Cross-modal hallucination mitigation addresses failures in integrating evidence across visual, audio, and textual modalities. Their mitigation approaches fall into two subcategories: (i) *temporal synchronization and interleaved encoding*; and (ii) *cross-modal alignment and contrastive tuning*.

**Temporal Synchronization and Interleaved Encoding** Fine-grained temporal alignment between audio and visual signals is essential for preventing misattribution of auditory cues. Early Vid-LLMs such as Video-LLaMA (Zhang et al., 2023b) processed modalities through largely independent adapters, resulting in weak audio exploitation. VideoLLaMA 2 (Cheng et al., 2024) replaced this with a Spatial-Temporal Convolution connector employing 3D convolutions and integrated a BEATs audio encoder, improving on standard video benchmarks but remaining coarse-grained beyond eight frames.

As an explicit temporal encoding method, OMCAT (Goel et al., 2024) introduces Rotary Time Embeddings (RoTE) encoding absolute timestamps as rotation angles into both audio and visual tokens, as many cross-modal hallucinations arise from the model's inability to determine which signal belongs to which moment. OMCAT nearly doubled VideoLLaMA 2's accuracy and outperformed AVicuna (Shu et al., 2023).

Along this direction, Dolphin (Guo et al., 2025d) addresses both spatial and temporal dimensions through a multi-scale audio-visual adapter and interleaved temporal merging, reducing hallucination rates by nearly 40% on speech-aware samples. Temporal synchronization is most effective when paired with source localization. Chen et al. (2025b) shifted to frame-level alignment through a streaming paradigm densely interleaving ASR word tokens with video frames based on precise timestamps, achieving state-of-the-art real-time commentary with $30\times$ latency reduction over Vid2Seq (Yang et al., 2023), though inheriting ASR errors. Ge et al. (2025) augmented captioning models with an auxiliary audio encoder and adaptive duration-based synchronization for multi-granularity temporally grounded captioning.

The central insight of these methods is: cross-modal hallucination often occurs asa sequencing problem rather as a representation problem. Models fail because they fuse modalities too early, too coarsely, or without a reliable notion of event order. A persistent challenge worth noting is how to handle temporally overlapping or weakly correlated audio-visual events.

**Cross-Modal Alignment** While temporal synchronization addresses *when* modalities should be aligned, cross-modal alignment addresses *what* should be aligned by enforcing representational consistency across video, audio, and text feature spaces.

Yang et al. (2024) proposed a self-supervised contrastive regularization framework that contrasts original and perturbed multimodal inputs with temporal order regularization, reducing reliance on superficial question-scene correlations beyond the invariant grounding of Li et al. (2022c). Bansal et al. (2023) leveraged contrast captions with temporal and action perturbations, improving temporal reasoning (Bagad et al., 2023; Momeni et al., 2023). Zhao et al. (2025b) replaced VideoCon's binary contrastive signal with a self-training hallucination correction objective, outperforming it by up to 17.9% on VELOCITI (Saravanan et al., 2025).

InternVideo2 (Wang et al., 2024i) introduced a progressive three-stage paradigm unifying masked video modeling, cross-modal contrastive learning across video-audio-speech-text, and next-token prediction within a 6B-parameter model. It goes beyond video-text contrastive scope (Li et al., 2023c) and achieves state-of-the-art across over 65 video and audio benchmarks. Li et al. (2024e) introduced object-aware adaptive-positivity contrastive learning aligning question-object and audio-object pairs individually on MUSIC-AVQA (Li et al., 2022a). Clearly, alignment has become more about constructing a shared multimodal space preserving motion, speech, and semantics.

In the causal reasoning line of work, Chen et al. (2025d) formulated a Structural Causal Model for multimodal deconfounding, employing front-door adjustment on visual features and back-door adjustment on linguistic features. Causal alignment is better suited to reducing unfaithful grounding than correlation-based methods alone, since a model may attend to aligned segments that are not the true basis for the answer. This reduced bias-induced errors over data-driven baselines (Xiao et al., 2024), demonstrating that causal interventions complement contrastive strategies in weakly supervised settings.

There are also architectural innovations for alignment. Liu et al. (2023a) proposed a cascaded Q-Former reducing severe hallucination rates from 34% to 20% on Video-CSR (Liu et al., 2023b). Additional efforts include mutual knowledge transfer (Weng & Li, 2022), bidirectional set-level alignment (Wang et al., 2023b), multi-grained correspondence learning (Wang et al., 2024j), codebook-based discretization (Zhu et al., 2025b), sparse masked pre-training (Lin et al., 2022), congruity matching (Zeng et al., 2023), and dual-task representations (Wang et al., 2024e). The most effective recent methods distinguish themselves by finer control over what is aligned: timestamps, objects, speech spans, event order, or causal evidence. Enforcing a shared, semantically consistent representation across modalities is a primary contributor, addressing what should be aligned rather than only when.

**Summary.** In the cross-modality category, temporal synchronization ensures correct chronological pairing, and cross-modal alignment enforces semantic consistency across feature spaces. Synchronization has evolved from independent fusion through timestamp encoding to fine-grained interleaving. Alignment spans contrastive regularization, multimodal pretraining, object-level grounding, and causal deconfounding. Fine-grained methods improve faithfulness but increase cost. Open challenges include robust reasoning over temporally overlapping or weakly correlated events, handling arbitrary modality combinations, and efficient scaling to long-form video.

# 7 Future Directions

We have mentioned many open challenges along the way, which limit the reliability of Vid-LLMs in real-world deployment. Here we highlight future directions we believe promising.

## 7.1 Hallucination Diagnosis and Root-Cause Attribution

While the taxonomy presented in this survey organizes hallucinations according to their underlying failure mechanisms, real-world hallucinations are often entangled and may simultaneously exhibit multiple symptoms. For example, a long-context memory failure may manifest as an object hallucination or temporal hallucination in the final output, and a temporal grounding error may subsequently induce incorrect causal reasoning.

An important future research direction is the development of systematic hallucination diagnosis frameworks that can attribute hallucinations to their root causes rather than only their observable symptoms. Such frameworks could analyze hallucination hierarchically along the Vid-LLM pipeline, and identify the earliest

failure mechanism responsible for the final unsupported output. When multiple failure mechanisms contribute simultaneously, future systems may support multi-label annotations or causal dependency graphs that explicitly represent interactions among hallucination types. This framework would improve the operational utility of hallucination taxonomies, enable more consistent benchmark annotation, and facilitate the selection of targeted mitigation strategies.

## 7.2 Out-of-Schema Strategies

Most mitigation methods assume that encountered entities, actions, and event structures fall within the training distribution (Maaz et al., 2023; Chen et al., 2024c). But real-world videos routinely present novel entities and previously unseen event configurations. When confronted with such new inputs, Vid-LLMs may project unfamiliar percepts onto the nearest learned prototype and generate fluent but fabricated descriptions.

Vid-LLMs trained through next-token prediction or contrastive alignment maximize likelihood over a fixed semantic space, encouraging confident predictions even when evidence is ambiguous (Li et al., 2024c; Lin et al., 2023). Promising avenues along this line include explicit uncertainty estimation over visual concepts, shifting from categorical labeling toward grounded open-vocabulary descriptions composing outputs from primitive visual attributes (Wasim et al., 2023; Munasinghe et al., 2024), and large-scale open-world data expansion incorporating diverse real-world streams and long-tail event combinations. Also, enabling Vid-LLMs to recognize the boundaries of their own knowledge would substantially reduce hallucinations that current closed-world methods leave unaddressed.

## 7.3 Temporal Causality and Action Dynamics via World-Modeling

Video-specific hallucinations stem not only from weak temporal modeling but also from insufficient physical-world understanding (Chang et al., 2025; Tang et al., 2025e). Current Vid-LLMs are believed to lack explicit representations of object permanence, motion continuity, and cause-effect transitions, relying instead on appearance cues and language priors (Zhan et al., 2025; Liu et al., 2024d). Future Vid-LLMs or Video World Models may explicitly model how the physical world evolves over time, learning latent world representations that maintain structured state information and predict state transitions under actions. Grounding reasoning in explicit world models could substantially reduce hallucinations related to action dynamics and false causal narratives.

# 8 Conclusion

Vid-LLMs and their capacities have rapidly advanced, but their reliability remains limited by hallucination, i.e., generating outputs not faithfully grounded in video evidence. Like and unlike hallucination in text or static images, video hallucination emerges from temporal reasoning, long-context information management, motion dynamics, and multimodal alignment.

This survey has presented a video-centric analysis of hallucination in Vid-LLMs. We introduce a unified taxonomy organizing video hallucination into six categories: temporal understanding, motion and action dynamics, long-context reasoning, object perception, causal reasoning, and cross-modal grounding. Then we systematically review mitigation techniques organized by the hallucinations they address.

Despite rapid progress, reliable video understanding requires further advances in hallucination-aware evaluation, open-world reasoning capabilities, and stronger modeling of the physical world. We hope this survey provides some help for researchers and practitioners working in the area of video understanding.

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

# A Distribution of Hallucination Mitigation Research

To complement our taxonomy and provide a broader view of the research landscape, we analyze the distribution of hallucination mitigation studies across years and different hallucination categories, following the taxonomy introduced in Section 3. We exclude papers whose mitigation targets could not be reliably categorized and omit 2026 statistics because the year is incomplete at the time of writing.

## A.1 Research Growth Over Time

Table 4: Number of hallucination mitigation papers by year and year-over-year growth rate.

| Year | Count | Growth Rate | Cumulative |
|------|-------|-------------|------------|
| 2022 | 105 | — | 105 |
| 2023 | 197 | +87.6% | 302 |
| 2024 | 302 | +53.3% | 604 |
| 2025 | 541 | +79.1% | 1,145 |

The overall volume of research has increased substantially, growing from 105 papers in 2022 to 541 papers in 2025, with a total of 1,145 papers across the four-year period, as shown in table 4. This more than five-fold increase reflects the rapid expansion of Vid-LLMs and the growing recognition of hallucination as a critical reliability challenge. The year-over-year growth rates consistently exceed 50%, with the highest growth of +87.6% occurring between 2022 and 2023, demonstrating the accelerating pace of research in this area.

The largest absolute increase occurs between 2024 and 2025, coinciding with the rapid emergence of large-scale video-language foundation models and long-video reasoning systems. This trend suggests that hallucination mitigation has evolved from a niche research topic into an important subfield of video understanding.

## A.2 Distribution Across Hallucination Categories

Table 5: Distribution of mitigation papers across hallucination categories (2022–2025).

| Hallucination Category | Percentage |
|------------------------|------------|
| Long-Context Video Understanding | 26.0% |
| Temporal Understanding | 21.3% |
| Cross-Modal | 20.4% |
| Object | 11.9% |
| Causal and Reasoning | 10.5% |
| Motion and Action Dynamics | 9.9% |

We summarize the percentage distribution of mitigation papers across hallucination categories from 2022–2025 in table 5, aggregated over the full four-year period. Three categories form the top tier of mitigation effort. Long-context video understanding hallucination attracts the greatest share (26.0%): the popularity of long-video applications has motivated substantial work on memory mechanisms, retrieval strategies, token compression, and frame selection, making the reliable scaling of Vid-LLMs to longer videos a central concern. Temporal understanding hallucination follows closely (21.3%), which is unsurprising given that temporal ordering, grounding, and event localization are fundamental requirements for nearly all video understanding tasks. Cross-modal hallucination accounts for a comparably large share (20.4%), reflecting sustained effort on aligning visual, audio, and textual streams—though, as we show in Section A.3, this aggregate figure is driven largely by early-period activity rather than current momentum.

The remaining three categories receive comparatively smaller but relatively balanced attention: object (11.9%), causal and reasoning (10.5%), and motion and action dynamics (9.9%) hallucination. Object and motion-and-action mitigation remain closely tied to perception and fine-grained spatiotemporal alignment, which are mature yet still active directions. Causal and reasoning hallucination, by contrast, is increasingly recognized as a major limitation of Vid-LLMs, yet relatively few mitigation approaches explicitly target

causal inference, counterfactual reasoning, or intention understanding. This gap suggests that reasoning-oriented hallucination mitigation remains an open and potentially high-impact research direction.

## A.3 Evolution of the Mitigation Research

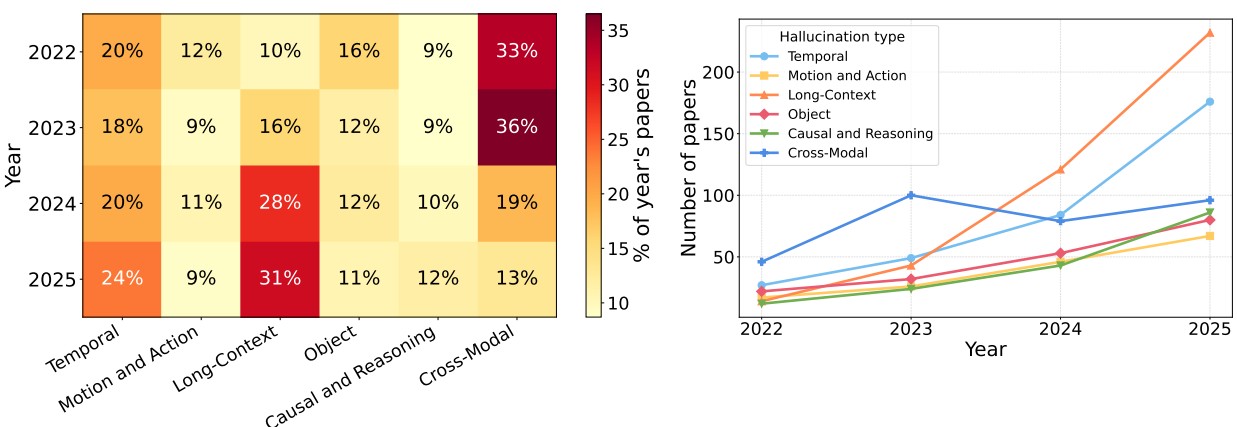

(a) Per-year share of each hallucination type.

(b) Per-year number of mitigation papers.

Figure 11: Evolution of mitigation research across the six hallucination types (2022–2025). (a) For each year, the percentage of that year's mitigation papers devoted to each hallucination type. (b) The absolute number of mitigation papers targeting each hallucination type per year.

The aggregate distribution above conceals substantial shifts in where mitigation effort has been directed over time. Figure 11 disentangles these dynamics from two complementary angles: the heatmap (Figure 11a) reports, for each year, the fraction of that year's mitigation papers assigned to each hallucination type, while the line chart (Figure 11b) reports the absolute number of mitigation papers per type.

The most striking pattern is the front-loaded trajectory of cross-modal hallucination mitigation. It was the dominant focus of the early period, comprising 33% of mitigation work in 2022 and 36% in 2023, when aligning visual, audio, and textual streams was the most pressing source of unfaithful video outputs. Its share then fell sharply to 19% in 2024 and 13% in 2025, and its absolute count peaked in 2023 (∼100 papers) before plateauing. This explains why its large four-year aggregate (20.4%) overstates its current momentum: the bulk of cross-modal mitigation work was carried out before 2024, after which the community's attention diversified.

In contrast, long-context video understanding hallucination mitigation is the clear breakout direction. Its within-year share climbs steadily from 10% (2022) to 31% (2025), and its absolute volume grows fastest of all categories—overtaking every other type by 2024 and reaching the highest count in 2025. This sustained acceleration mirrors the rise of long-video applications and the recognition that memory, retrieval, and token-compression failures are a primary reliability bottleneck. Temporal understanding hallucination mitigation shows a similarly healthy trajectory, holding a consistently high share (18–24%) while growing to the second-largest absolute volume, confirming it as a stable, core concern rather than a passing trend.

The remaining types—object, motion and action dynamics, and causal and reasoning hallucination—occupy a lower band of per-year shares (roughly 9–16%) throughout the period, with steady but more modest absolute growth. A notable exception is causal and reasoning hallucination mitigation, whose share edges up to 12% in 2025 alongside a sharp jump in absolute count, hinting that reasoning-oriented mitigation is beginning to attract the attention its importance warrants. Overall, the evolution reveals a field that has migrated from an early emphasis on cross-modal alignment toward long-context and temporal reliability, while reasoning-centric mitigation remains comparatively underexplored.

# B   Limitations

**Scope and coverage.**   The survey is deliberately restricted to hallucination in Vid-LLMs and related papers published between 2022 and 2026 (Section 2.3). It does not cover image-only hallucination, non-LLM video models such as pure detection or tracking systems, or hallucination in video generation, and its coverage of rapidly released or proprietary models is necessarily incomplete given the pace of the field. We view these boundaries as scope choices that keep the survey coherent around a single phenomenon rather than as gaps in the topic we set out to characterize; the excluded areas are nonetheless natural directions for complementary surveys, and some of the mechanisms we discuss are likely to transfer to those settings.

**No empirical re-benchmarking.**   Although we compare methods on the same benchmark, our analysis relies on the results reported in the original works rather than on a unified, controlled re-evaluation. Therefore, the underlying experimental settings—including the prompt template, frame-sampling strategy, and decoding configuration—may not be identical across the compared methods. In most cases we expect these differences to have only a limited effect on the overall trends and relative rankings. Nonetheless, a unified re-benchmarking protocol that fixes every setting would allow each method to be evaluated formally and strictly under exactly the same conditions, yielding more rigorous head-to-head comparisons. Carrying out such a controlled re-evaluation across all surveyed methods is beyond the scope of this survey and remains a valuable direction for future work.

**Domain breadth.**   Our analysis centers on general-purpose video understanding. Specialized domains such as egocentric, medical, autonomous-driving, and surveillance video are discussed only where they intersect the proposed taxonomy, and we do not attempt to exhaustively characterize the domain-specific hallucination patterns, benchmarks, and mitigation techniques that each of these areas may require. Such domains often introduce distinct priors, safety requirements, and failure costs—for instance, a hallucinated object in a driving or clinical setting carries far higher stakes than in generic video—which may demand specialized evaluation criteria and mitigation strategies that fall outside our general-purpose treatment. We expect the taxonomy to provide a useful starting point in these settings, but adapting it to each domain remains future work.

**Root-cause taxonomy vs. symptom-level evaluation.**   Our taxonomy organizes hallucinations by their underlying failure mechanism rather than their surface symptom, on the rationale that mitigation is best designed against mechanisms. This is a proposed organizing principle rather than a validated diagnostic procedure: a single output may exhibit several entangled symptoms, and reliably attributing one to its originating mechanism remains an open problem, so the benchmarks we survey operate at the symptom level and we can assess mechanism-targeted mitigation only indirectly. The taxonomy nonetheless remains useful, because entangled symptoms can usually be narrowed to a small set of separable candidate mechanisms at distinct pipeline stages, each addressed by a different mitigation family—enough to guide the selection of targeted mitigations. Even so, more reliable, root-cause-oriented evaluation would let these mechanism-level claims be verified directly rather than inferred from symptom-level benchmarks, and would further strengthen the work as attribution frameworks (Section 7.1) mature.

