# OpenReview forum: "A Survey on Hallucination in Video Understanding: Taxonomy, Causes, and Mitigation Techniques"
_TMLR — Decision pending for TMLR_

### Review · Reviewer_msJJ · 2026-05-13

**Summary Of Contributions:**

This paper surveys hallucination in Video Large Language Models (Vid-LLMs), where models generate answers or descriptions that are not faithfully grounded in the input video. It argues that video hallucination deserves focused treatment because video understanding adds temporal order, motion, long-context memory, causal reasoning, and cross-modal alignment challenges beyond those seen in text-only or image-based models.

The paper's main contribution is a six-part taxonomy of Vid-LLM hallucination: temporal understanding, motion and action dynamics, long-context video understanding, object hallucination, causal and reasoning hallucination, and cross-modal hallucination. The taxonomy is illustrated with concrete video scenarios and is paired with a review of mitigation methods, including preference optimization, synthetic supervision, structured representations, long-context evidence management, and inference-time interventions.

Overall, the survey is timely and useful. Its strongest contribution is the organization of a fragmented literature into a video-specific taxonomy linked to mitigation strategies. However, the current draft would benefit from stronger evidence organization, clearer taxonomy boundaries, and a more systematic treatment of benchmarks and metrics.

**Additional Comments:**

This is a timely and potentially valuable survey. The taxonomy and mitigation mapping are useful, but the current draft needs stronger survey infrastructure before acceptance: benchmarks, metrics, comparison tables, clearer category boundaries, and more careful calibration of empirical claims. With these revisions, the paper could become a strong reference for Vid-LLM hallucination research.

**Audience:**

Yes

**Audience Explanation:**

This topic is clearly relevant to TMLR. Researchers in multimodal learning, NLP, computer vision, model evaluation, and AI safety would benefit from a structured account of hallucination in Vid-LLMs.

**Broader Impact Concerns:**

No concerns

**Claims And Evidence:**

No

**Claims Explanation:**

The paper makes a convincing case that hallucination in Vid-LLMs is an important and distinct reliability problem. The examples and taxonomy are intuitive, and the literature coverage is broad. The link between hallucination types and mitigation families is also valuable.

That said, several claims are not yet supported as clearly as they should be.
First, the paper claims to fill a gap left by prior surveys, but it does not provide a direct comparison showing what prior surveys cover and what this survey uniquely adds. Since the contribution is primarily organizational, a comparative scope table is important. The introduction (§1) cites at least six video understanding surveys (Nguyen et al. 2024; Tang et al. 2025c; Zhong et al. 2022; Zhou et al. 2024; Madan et al. 2024; Zou et al. 2024) and two cross-modal hallucination surveys (Sahoo et al. 2024; Xia et al. 2025) but addresses them in a single paragraph without a structured comparison. Since the contribution is primarily organizational, a comparative scope table is important.

Second, the taxonomy is useful but not fully operational. Some categories naturally overlap: long-context failures (§3.3) can produce temporal or object hallucinations; motion/action errors (§3.2) and temporal errors (§3.1) both involve event timing and are separated only by granularity; and false-premise text-video hallucination (§3.6) and false causality (§3.5) both describe the model overriding visual evidence with linguistic priors, making their boundary unclear. Section 2.3 explicitly acknowledges that "perceptual, temporal, and reasoning errors are possibly entangled," yet §3 provides no decision criteria for borderline cases.  The paper provides no procedure for classifying such cases.

Third, the mitigation evidence is difficult to evaluate. Section 5 reports all results in prose across heterogeneous benchmarks. For example, §5.1 reports Time-R1 reducing hallucination rates by 10–15% on Charades-STA and TimeWarp reducing hallucination by 7.5% on temporal benchmarks, while §5.2 reports MASH-VLM reducing hallucination by 12–17% on UNSCENE. These figures use different metrics, baselines, and datasets, making cross-method comparison impossible. Without structured tables, it is hard to tell which methods address which hallucination types, which results directly measure hallucination, and which merely improve general task accuracy.

**Requested Changes:**

1.  Add a benchmarks and metrics section or appendix mapping hallucination types to relevant datasets, metrics, and evaluation limitations. Section 6.1 sketches the right requirements (negative labels, grounding-aware metrics, hallucination-type diversity) but frames them only as future work; that material should be reorganized into a current-state evaluation review.
2.  Add a comparison with prior surveys to substantiate the claimed gap. The §1 introduction lists at least six video understanding surveys and two cross-modal hallucination surveys, but engages with them only in prose. A table comparing scope, taxonomy depth, mitigation coverage, and video-specificity across these works would substantiate the claim that this survey uniquely fills a gap.
3. Clean up the bibliography. Multiple clear duplicates are present for example Chang et al. (2025a) and (2025b) are the same preprint (arXiv:2512.04356);
4. Add a scope and limitations section clarifying what the survey does and does not cover.

---

> ### Author Response · Authors · 2026-06-26
>
> We thank the reviewer for the careful and constructive feedback. We appreciate the reviewer’s positive comments that the survey is timely and useful, makes a convincing case for Vid-LLM hallucination as a distinct reliability problem, and provides a valuable organization of a fragmented literature into a video-specific taxonomy linked to mitigation strategies. We also thank the reviewer for identifying important concerns regarding evidence organization, taxonomy boundaries, benchmark and metric coverage, comparison with prior surveys, and bibliography quality. We have revised the manuscript to address these concerns, and we provide the detailed responses below.

---

> > ### Author Response · Authors · 2026-06-26
> >
> > ## [RC 1] Add a benchmarks and metrics section mapping hallucination types to relevant dataset.
> >
> > We have added a new current-state evaluation review, **§4 "Benchmarks and Evaluation".** The section is organized by the hallucination taxonomy and is anchored by **Table 2,** which lists, for each benchmark, its primary hallucination category, year, evaluation format (e.g., grounding/IoU, MC QA, binary QA, entailment, captioning), and key evaluation focus. The per-category text discusses the metrics and the specific evaluation limitations of each benchmark (e.g., negative/adversarial labels in VideoHallucer, conflicting-video construction in TempCompass, needle-in-a-haystack design in NIAVH), and notes where evaluation methodology lags mitigation research—most visibly the scarcity of dedicated cross-modal hallucination benchmarks **(§4.6),** which we connect to the cross-modal mitigation gap in **§5.6.** The forward-looking requirements that were previously sketched as future work (negative labels, grounding-aware metrics, type diversity) are retained but now framed against this current-state review.
> >
> > ## [RC 2] Add a structured comparison with prior surveys.
> > We have replaced the single-paragraph prose treatment with a structured comparison. We've added **Table 1 (§1)** now to compare this survey against the six video-understanding surveys (Zhong et al. 2022; Madan et al. 2024; Nguyen et al. 2024; Zhou et al. 2024; Tang et al. 2025; Zou et al. 2024) and the two cross-modal hallucination surveys (Sahoo et al. 2024; Xia et al. 2025) along two grouped axes: General Video-Understanding Coverage (models & architectures, tasks & benchmarks, challenges & future) and Hallucination-Specific Coverage (hallucination focus, video-specific definition, taxonomy, benchmarks, mitigation-by-hallucination-type), using a ✓/✗/∘ legend. The surrounding text in **§1** now engages each group individually—noting that the video-understanding surveys mention hallucination only in passing (∘ for Tang et al. and Nguyen et al.) and that the cross-modal surveys apply modality-uniform taxonomies without video-specific definitions or mitigation-by-type organization. This substantiates the claimed gap as a structured comparison rather than prose.
> >
> > ## [RC 3] Clean up the bibliography and remove duplicate references.
> > We have cleaned up the bibliography. The specific duplicate the reviewer identified—Chang et al. (2025a) and (2025b), both arXiv:2512.04356—has been consolidated into a single entry. We additionally performed a systematic deduplication pass over the entire bibliography: across all 1,214 entries there are now **no duplicate citation keys and no duplicate arXiv identifiers.** We thank the reviewer for catching this.

---

> > > ### Author Response · Authors · 2026-06-26
> > >
> > > ## [RC 4] Add a scope and limitations section clarifying what the survey covers and does not cover.
> > >
> > > We have clarified the survey’s scope and limitations in two complementary places.
> > >
> > > First, the new subsection **§2.3 "Literature Collection Methodology"** defines what the survey covers: the time range (2022–2026), the paper selection process (keyword search and citation analysis), and explicit inclusion and exclusion criteria. In particular, we include works that analyze hallucination phenomena, introduce hallucination-related benchmarks or evaluation protocols, propose mitigation techniques, or address closely related reliability challenges (temporal grounding, long-context reasoning, multimodal alignment); and we exclude image-only hallucination studies without video relevance, works unrelated to reliability or grounding in video, and duplicate versions of the same work. This makes the boundary of the surveyed literature explicit.
> > >
> > > Second, we add an explicit **Appendix B "Limitations"** stating what the survey does **not** cover, with three parts: (i) Scope and coverage—it does not cover image-only hallucination, non-LLM video models (e.g., pure detection/tracking), or hallucination in video generation; (ii) No empirical re-benchmarking—we report results as published rather than running a unified controlled re-evaluation; and (iii) Domain breadth—specialized domains (egocentric, medical, autonomous-driving, surveillance) are discussed only where they intersect the taxonomy.
> > >
> > > ## [RC 5] Report the results across consistent benchmarks.
> > > We agree that comparing numbers drawn from different benchmarks, baselines, and metrics is misleading, and we address this in two coordinated steps.
> > >
> > > First, we add a new section, **§4 "Benchmarks and Evaluation",** that shows, for each of the six hallucination types, the benchmarks used to evaluate it (organized per type in **§4.1–§4.6** and summarized in **Table 2).** This makes the evaluation landscape explicit and clarifies, for each hallucination type, which datasets and evaluation formats are standard.
> > >
> > > Second, building on this section, we update the mitigation techniques in **§6** so that, when introducing methods with same kind, their results are reported on the **same benchmark** within each comparison rather than on heterogeneous ones. For example, Sparrow and TimeWarp are now both reported on TempCompass, so their reported gains can be read against the same reference. A few non-comparable "hallucination-rate reduction" figures that could not be tied to a common protocol were removed. Anchoring each method’s reported results to the benchmarks established in **§4** makes cross-method comparison within each hallucination type consistent and clear.

---

> > > > ### Author Response · Authors · 2026-06-26
> > > >
> > > > ## [Concern 1] The hallucination taxonomy is useful but not fully operational. Some categories naturally overlap.
> > > >
> > > > We thank the reviewer for identifying these specific overlaps. **We acknowledge that different hallucination types can co-occur and entangle at the output level** —a single model response may simultaneously exhibit multiple hallucination symptoms. However, we argue that **the taxonomy remains valuable** because these entangled symptoms trace back to distinct root causes at different pipeline stages, and each root cause demands a different family of mitigation techniques.
> > > >
> > > > As the reviewer mentioned, long-context failures **(§3.3)** producing temporal or object hallucinations. In practice, a long-context memory failure may simultaneously produce a temporal ordering error and an object attribute error in the same model output. These hallucinations are indeed entangled at the symptom level. However, the same temporal ordering error has fundamentally different origins depending on context: in a short clip, it arises from weak temporal encoding; in an hour-long video, it arises from memory degradation under context compression. These different root causes demand different interventions—temporal encoding improvements cannot fix a memory retrieval failure, and memory optimization cannot fix a fundamentally weak temporal representation. The taxonomy separates by root cause so that, even when symptoms entangle, researchers can trace each failure back to its origin and select the appropriate mitigation.
> > > >
> > > > As the reviewer pointed out, motion/action errors **(§3.2)** and temporal errors **(§3.1)** sharing event timing. These two categories can certainly co-occur. A model may simultaneously misorder events and mischaracterize the motion within those events. Yet their root causes are distinct: temporal hallucination originates from failures in chronological structure encoding, while motion hallucination originates from failures in dynamics representation.  Even when both errors appear together, they trace to different pipeline components: temporal mitigations target timestamp encoding and event ordering supervision, while motion mitigations target trajectory tracking, and physics-informed modeling. The taxonomy facilitates targeted diagnosis of root causes and the development of corresponding mitigation strategies.
> > > >
> > > > As the reviewer highlighted, false-premise text-video hallucination **(§3.6)** and false causality **(§3.5)** both overriding visual evidence with linguistic priors. These can entangle when a misleading prompt triggers a fabricated causal explanation. However, their root causes differ in origin and mechanism: false-premise hallucination is rooted in the model's inability to reject externally imposed false assumptions in the query, while false causality is rooted in the model's tendency to internally generate spurious causal links from learned stereotypical correlations. The first is a query-robustness failure; the second is a generation-side reasoning failure. Correspondingly, mitigating false-premise hallucination requires premise verification and query-grounding mechanisms, while mitigating false causality requires grounded causal reasoning and suppression of prior-driven confabulation. Even when both appear in the same output, they require different interventions because their root causes reside at different stages.
> > > >
> > > > In conclusion, the taxonomy does not claim that hallucination categories are mutually exclusive at the symptom level—real-world failures are often entangled. Rather, it provides a root-cause-oriented decomposition where each category corresponds to a distinct failure mechanism at a specific pipeline stage. This is precisely what makes the taxonomy operationally useful: it maps entangled symptoms back to separable root causes, each addressable by a targeted family of mitigations.
> > > >
> > > > That said, we agree with the reviewer on an important operational point. Defining categories by root cause is at present a proposed organizing principle rather than a fully operational diagnostic procedure—we cannot yet always attribute an observed output reliably to a single originating mechanism. The sole reason we nonetheless adopt a root-cause organization is that mitigation is best designed against mechanisms, not symptoms. In practice, evaluation, annotation, and benchmarks therefore remain symptom-level, and we can assess mechanism-targeted mitigation only indirectly through symptom-level benchmarks. We have made this explicit in the revision: we record it in our Limitations and add Hallucination Diagnosis and Root-Cause Attribution to Future Work **(§7.1).** We expect the proposed mitigation taxonomy to translate into measurable gains on root-cause-oriented benchmarks once attribution frameworks mature, which we leave to future validation.

---

### Review · Reviewer_yLZF · 2026-05-21

**Summary Of Contributions:**

In this work, the authors propose a survey on hallucination in Vid-LLMs. It provides a taxonomy of hallucination types, including temporal understanding hallucination, motion and action dynamics hallucination, long-context video understanding hallucination, object hallucination, causal and reasoning hallucination, and cross-modal hallucination. The authors also organize mitigation techniques for each category and discuss future directions such as hallucination-specific benchmarks, out-of-schema reasoning, and world-modeling for video understanding.

Overall, the manuscript is easy to follow and covers a timely topic. I think the paper can be useful for researchers working on video-language models and hallucination. However, I have some concerns about the survey methodology, the definition of hallucination, and the level of critical analysis in the paper.

**Audience:**

Yes

**Audience Explanation:**

The topic is timely and relevant to the TMLR audience, especially with the increasing use of video-language models and multimodal LLMs. The paper can be useful as an entry point for researchers who want to understand hallucination in video understanding.

**Claims And Evidence:**

No

**Claims Explanation:**

My overall assessment is that the paper covers many relevant works and the proposed organization is intuitive. However, I have some concerns that I would like to raise here:

1. The authors claim that this is a comprehensive and systematic review. However, I could not find a clear description of the survey protocol. How were the papers selected? What was the time range? What were the inclusion and exclusion criteria?

2. The definition of hallucination is also sometimes too broad. Several examples seem to be general video-understanding errors, such as wrong counting, wrong temporal grounding, wrong motion direction, or wrong attribute prediction. These are important errors, but it is not always clear why they should all be called hallucination rather than standard perception, localization, or reasoning errors. I think the authors should more clearly separate hallucination from general prediction errors.

3. The mitigation section is broad, but it sometimes reads like a list of papers rather than a critical synthesis. Many methods are summarized one after another, but it is hard to understand which method family is most useful under which condition. I think a table comparing methods by hallucination type, benchmark, metric, required supervision, computational cost, and main limitation would make the survey much stronger.

4. Also, while the taxonomy is helpful, it is not clear which hallucination categories have attracted more attention in the literature and which ones are relatively overlooked. For example, are most existing works focused on temporal grounding and object hallucination, while cross-modal or causal/reasoning hallucinations are less studied? Since the paper already organizes hallucination into six categories and mitigation methods into corresponding groups, I think a distribution analysis would make the survey much more informative. I recommend that the authors provide a chart showing the number of papers reporting each hallucination type and the number of papers proposing solutions for each category over recent years. This would help readers understand the trend of the field, identify saturated directions, and see which hallucination types still need more research attention.

**Requested Changes:**

In my answer above, I explained what my concerns are and how to address them. I also mention my major suggestions here:

1. A clear survey methodology section, including search keywords, databases/venues, time range, inclusion/exclusion criteria, and the number of papers covered.

2. A table that maps each surveyed paper to hallucination category, task, benchmark, metric, mitigation strategy, and limitations.

3. Clarify the distinction between hallucination and general video-understanding error.

4. A distribution chart or timeline analysis showing how many papers focus on each hallucination category and how many mitigation methods have been proposed for each category over recent years.

---

> ### Author Response · Authors · 2026-06-26
>
> We thank the reviewer for the careful and constructive feedback. We appreciate the reviewer’s positive comments that the manuscript is easy to follow, addresses a timely topic, and can serve as a useful entry point for researchers studying video-language models and hallucination. We also thank the reviewer for raising important concerns about the survey methodology, the definition of hallucination, the need for deeper synthesis, and the lack of distribution analysis across hallucination categories. We have revised the manuscript to address these concerns, and we provide the detailed responses below.

---

> > ### Author Response · Authors · 2026-06-26
> >
> > ## [RC 1] Add a clear survey methodology section.
> >
> > We agree and have added a dedicated subsection, **§2.3 "Literature Collection Methodology",** describing the full survey protocol: the **time range** (2022–2026) with an explicit rationale for the 2022 start (the shift from task-specific discriminative video models to generative, instruction-following Vid-LLMs, beginning with Flamingo in 2022 and the wave of dedicated Vid-LLMs in 2023); the **search process** (keyword-based search plus citation analysis, with benchmarks/datasets collected separately); and the **inclusion/exclusion criteria** (we include works that analyze hallucination phenomena, introduce hallucination-related benchmarks or evaluation protocols, propose mitigation techniques, or address closely related reliability challenges such as temporal grounding, long-context reasoning, and multimodal alignment; we exclude image-only hallucination studies without video relevance, works unrelated to reliability/grounding in video, and duplicate versions of the same work). The **number of papers covered and relevant distribution analysis** is reported in **Appendix A.**

---

> ### Author Response · Authors · 2026-06-26
>
> ## [RC 2] Provide a comparison table mapping hallucination category, benchmark, and mitigation strategy.
>
> We thank the reviewer for this suggestion. To connect the three axes—hallucination type, evaluation benchmark, and mitigation technique—into a single coherent mapping, we made three additions to the revised manuscript:
> ### 1. Benchmarks for each hallucination type — new Section 4 (Table 2).
> We added a "Benchmarks and Evaluation" section that organizes representative benchmarks under each of the six hallucination categories and lists them in **Table 2,** reporting for each benchmark its primary hallucination category, year, evaluation format, and key evaluation focus. This makes explicit, for every hallucination type, which datasets and metrics are used to measure it, so a reader can see how each failure mode is evaluated. It also enables a more structured introduction of the mitigation techniques: with the standard benchmark for each hallucination type established in **§4**, we report mitigation methods on consistent benchmarks when introducing them in **§6**, so their reported gains can be read against a common reference within each hallucination type.
> ### 2. Effectiveness of mitigation foundations for each hallucination type — new Section 5.6 (Table 3).
> We added **§5.6,** which rates how strongly each of the five mitigation foundations addresses each hallucination type in **Table 3.** These ratings are derived from a qualitative assessment of the surveyed methods in the main text; for each method family and hallucination type, they are based on two factors—how prevalently that family is used to address the type, and how effective its reported results are. Beyond mapping techniques to types, this offers a practical recommendation of which method family to prioritize for a given hallucination type, and clarifies why some families are more effective than others.
> ### 3. Outline of mitigation techniques for each hallucination type — Figure 5.
> **Figure 5** presents the taxonomy/outline of concrete mitigation techniques grouped under each hallucination category, so a reader can see at a glance which families of methods target which hallucination type and trace down to the specific representative methods within each.
>
> Together, these three components form an end-to-end mapping across hallucination type - benchmark - mitigation technique: **§4/Table 2** links each type to the benchmarks that measure it, **§5.6/Table 3** links each type to the most effective mitigation foundations with supporting evidence, and **§6/Figure 5** lays out the concrete techniques per type. Beyond a static summary, this lets a reader trace any hallucination type to how it is evaluated and how it is best mitigated, providing actionable guidance for selecting mitigation strategies rather than only a catalog of papers.

---

> > ### Author Response · Authors · 2026-06-26
> >
> > ## [RC 3] Clarify the distinction between hallucination and general video-understanding errors.
> >
> > We thank the reviewer for this important observation. Our survey adopts a **grounding-based definition of hallucination:** a model output is considered hallucinated whenever it is unsupported by or inconsistent with the underlying video evidence. Under this definition, errors such as temporal grounding mistakes, motion-direction errors, and unsupported causal conclusions are all forms of hallucination because the generated output does not faithfully reflect the video content.
> >
> > We agree that these failures can also be viewed as traditional perception, localization, or reasoning errors. However, **our goal is to provide a unified reliability-oriented framework for Vid-LLMs,** which has been introduced in section **§1.** While general taxonomies classify errors by which subsystem fails, hallucination taxonomies classify errors by how the final output departs from the available evidence. This perspective is particularly useful because many mitigation methods—including temporal grounding supervision, retrieval-based memory mechanisms, and preference optimization—aim to improve grounding fidelity regardless of whether the underlying failure originates from perception, localization, memory, or reasoning.

---

> > > ### Author Response · Authors · 2026-06-26
> > >
> > > ## [RC 4] Add a distribution and timeline analysis of papers across hallucination categories and mitigation techniques.
> > >
> > > We thank the reviewer for this valuable suggestion. We have added a new **Appendix A, "Distribution of Hallucination Mitigation Research",** with three complementary analyses—covering both the distribution across categories and the timeline of how that distribution evolves:
> > >
> > > - **Research growth over time (§A.1, Table 4):** we analyze the total number of mitigation papers per year. The corpus grows more than five-fold from 2022 to 2025, confirming hallucination mitigation as a rapidly expanding subfield.
> > > - **Aggregate distribution across categories (§A.2, Table 5):** we analyze the share of mitigation work per hallucination type over the full period. Long-context, temporal, and cross-modal form the top tier, while object, causal and reasoning, and motion receive smaller, relatively balanced attention.
> > > - **Timeline / evolution across categories (§A.3):** we analyze how each category's share and volume change year by year. The main finding is that attention has migrated from an early focus on cross-modal alignment toward long-context and temporal reliability, while causal and reasoning remains the most under-explored throughout.
> > >
> > > Together, these analyses give readers a clear view of which hallucination categories are saturated, which are growing, and which remain under-explored, directly addressing the reviewer's request for a distribution and timeline analysis across hallucination categories.

---

> > > > ### Author Response · Authors · 2026-06-26
> > > >
> > > > ## [RC 5] Reorganize the mitigation section to emphasize the conditions under which each technique is useful and the trade-offs among different methods.
> > > >
> > > > We have substantially reorganized the mitigation section to emphasize synthesis rather than enumeration. In the revised **§5,** before reviewing methods by hallucination category, we first distill the literature into five foundational method families—RL and preference optimization, supervision augmentation, structured representations, long-context information management, and inference-time intervention—organized according to where they intervene in the Vid-LLM pipeline. We then add a new synthesis section, **§5.6, “Effectiveness on Different Hallucination Types,”** which cross-cuts these method families against the six hallucination types in **Table 3.** This 5×6 matrix rates the effectiveness of each family for each hallucination type as strong, moderate, limited, or negligible. These ratings are derived from a qualitative assessment of the surveyed methods in the main text, based on two factors: how prevalently each family is used to address a type, and how effective its reported results are.
> > > >
> > > > The accompanying discussion highlights the main patterns across methods and hallucination types. Structured representations emerge as the most broadly used remedy; RL is widely applicable and particularly effective for temporal and causal-reasoning hallucinations; long-context information management is more specialized and rarely transferred to other hallucination types; synthetic supervision mainly benefits temporal hallucination; inference-time intervention is most useful for causal/reasoning hallucination; and cross-modal hallucination remains comparatively under-addressed, with existing solutions relying primarily on representation-level alignment.
> > > >
> > > > In **§6,** we further add explicit trade-off statements to each method paragraph, explaining how strongly each method family contributes to the target hallucination type and why, while connecting the discussion back to the **Table 3** ratings. For example, we clarify why verifiable rewards such as tIoU make RL one of the strongest levers for temporal hallucination, but only a secondary lever for object hallucination. Together, these revisions make the mitigation section a structured account of when each method family is most useful, what trade-offs it involves, and why some techniques are more effective than others for particular hallucination types.

---

### Review · Reviewer_4uto · 2026-06-13

**Summary Of Contributions:**

This paper surveys hallucination in video large language models, with a focus on video-specific failure modes and mitigation techniques. The main contribution is a taxonomy that organizes video hallucination into six categories: temporal understanding, motion and action dynamics, long-context video understanding, object hallucination, causal and reasoning hallucination, and cross-modal hallucination. The paper also reviews mitigation methods for each category, including reinforcement learning, preference optimization, structured representations, memory and retrieval methods, frame sampling, graph-based reasoning, object-centric grounding, and cross-modal alignment. A key strength of the paper is that it treats video hallucination as a distinct problem rather than simply extending image or text hallucination. However, the main weaknesses are that the survey methodology is not clearly described, the taxonomy boundaries sometimes overlap, and the mitigation section often reads more like a list of papers than a critical synthesis.

**Audience:**

Yes

**Audience Explanation:**

Yes, the paper would be interesting to part of the TMLR audience, especially researchers working on multimodal learning, video-language models, trustworthy AI, hallucination evaluation, and long-context reasoning.

**Claims And Evidence:**

No

**Claims Explanation:**

The paper supports its main claims with a large number of citations and gives useful examples for different hallucination types. The figures and taxonomy make the problem easier to understand, and the paper covers many recent mitigation directions. However, the evidence is not always presented in a fully convincing way. The paper does not explain how the surveyed works were selected, what search process was used, or what inclusion and exclusion criteria were followed. In addition, many reported results come from different papers, models, datasets, and evaluation setups, but the paper does not always discuss whether these numbers are directly comparable. The taxonomy is reasonable, but the paper should better justify the boundaries between categories, especially between temporal, motion, long-context, and causal hallucinations. Overall, the claims are plausible and mostly supported by references, but the paper needs stronger methodology and more critical analysis to make the evidence clearer and more convincing.

**Requested Changes:**

The authors should add a clear survey methodology section describing the search process, time range, venues or sources considered, and inclusion/exclusion criteria. They should also provide comparison tables summarizing hallucination types, representative benchmarks, metrics, mitigation methods, required supervision, model access, inference cost, and limitations. The taxonomy should be refined with clearer category boundaries and examples of borderline cases. The mitigation section should be reorganized to emphasize higher-level insights and trade-offs rather than listing many papers sequentially. The authors should also be more careful when comparing results across different benchmarks and setups. Finally, the paper needs substantial writing polish to remove informal phrasing, grammar issues, repetition, and unclear sentences.

---

> ### Author Response · Authors · 2026-06-26
>
> We thank the reviewer for the careful and constructive feedback. We appreciate the reviewer’s positive comments on the paper’s focus on video-specific hallucination, the usefulness of the proposed taxonomy, and the plausibility of our claims as supported by relevant references. We also thank the reviewer for raising important concerns about the survey methodology, taxonomy boundaries, and comparability of results. We have revised the manuscript to address these concerns, and we provide the detailed responses below.

---

> ### Author Response · Authors · 2026-06-26
>
> ## [RC 1] Add a survey methodology section describing the literature collection rules.
> We agree that the literature collection process was not sufficiently described in the original manuscript, and we have addressed this concern directly.
>
> We have added a dedicated subsection, **§2.3 "Literature Collection Methodology"**, that explicitly describes the survey protocol. It clarifies (i) the time range covered by the survey (2022–2026), together with an explicit justification for choosing 2022 as the starting year—it marks the transition from the task-specific discriminative paradigm (closed-set action recognition, temporal localization, captioning) toward the generative, instruction-following Vid-LLMs on which open-ended hallucination arises, beginning with Flamingo (2022) and the rapid wave of dedicated Vid-LLMs in 2023; (ii) the **paper selection process**, based on keyword-based search and citation analysis, plus the separate collection of the benchmarks/datasets these works rely on; and (iii) the **inclusion and exclusion criteria.** Specifically, we include papers that (1) analyze hallucination phenomena in video understanding, (2) introduce hallucination-related benchmarks, datasets, or evaluation protocols, (3) propose mitigation techniques for hallucination and grounding failures, or (4) address closely related reliability challenges such as temporal grounding, long-context reasoning, and multimodal alignment. We exclude image-only hallucination studies without video-specific relevance, works unrelated to reliability or grounding in video understanding, and duplicate versions of the same work.
>
> We also note that the 2022 starting point is empirically conservative: as reported in **Appendix A (Table 4),** the number of hallucination-related papers in our corpus grows from roughly one hundred in 2022 to more than five times that figure by 2025, confirming that the survey captures the field essentially from its inception.

---

> ### Author Response · Authors · 2026-06-26
>
> ## [RC 2] Provide comparison tables summarizing hallucination types, representative benchmarks, and mitigation methods.
>
> We thank the reviewer for this suggestion. To connect the three axes—hallucination type, evaluation benchmark, and mitigation technique—into a single coherent mapping, we made three additions to the revised manuscript:
> ### 1. Benchmarks for each hallucination type — new Section 4 (Table 2).
> We added a "Benchmarks and Evaluation" section that organizes representative benchmarks under each of the six hallucination categories and lists them in **Table 2,** reporting for each benchmark its primary hallucination category, year, evaluation format, and key evaluation focus. This makes explicit, for every hallucination type, which datasets and metrics are used to measure it, so a reader can see how each failure mode is evaluated. It also enables a more structured introduction of the mitigation techniques: with the standard benchmark for each hallucination type established in **§4**, we report mitigation methods on consistent benchmarks when introducing them in **§6**, so their reported gains can be read against a common reference within each hallucination type.
> ### 2. Effectiveness of mitigation foundations for each hallucination type — new Section 5.6 (Table 3).
> We added **§5.6,** which rates how strongly each of the five mitigation foundations addresses each hallucination type in **Table 3.** These ratings are derived from a qualitative assessment of the surveyed methods in the main text; for each method family and hallucination type, they are based on two factors—how prevalently that family is used to address the type, and how effective its reported results are. Beyond mapping techniques to types, this offers a practical recommendation of which method family to prioritize for a given hallucination type, and clarifies why some families are more effective than others.
> ### 3. Outline of mitigation techniques for each hallucination type — Figure 5.
> **Figure 5** presents the taxonomy/outline of concrete mitigation techniques grouped under each hallucination category, so a reader can see at a glance which families of methods target which hallucination type and trace down to the specific representative methods within each.
>
> Together, these three components form an end-to-end mapping across hallucination type - benchmark - mitigation technique: **§4/Table 2** links each type to the benchmarks that measure it, **§5.6/Table 3** links each type to the most effective mitigation foundations with supporting evidence, and **§6/Figure 5** lays out the concrete techniques per type. Beyond a static summary, this lets a reader trace any hallucination type to how it is evaluated and how it is best mitigated, providing actionable guidance for selecting mitigation strategies rather than only a catalog of papers.

---

> > ### Author Response · Authors · 2026-06-26
> >
> > ## [RC 3] Provide clearer category boundaries and borderline cases.
> > We thank the reviewer for this point and have strengthened the treatment of category boundaries in three ways.
> >
> > We acknowledge that the current survey does not provide a formal procedure for uniquely attributing every borderline case to a single hallucination category. We believe this **remains a challenging open problem** because hallucinations in Vid-LLMs often emerge through error propagation across multiple stages of the pipeline. For example, a long-context memory failure may ultimately manifest as an object or temporal hallucination in the generated response. Since the final output typically reveals only the observable symptom rather than the complete causal chain, different underlying failure mechanisms can produce highly similar hallucination behaviors. To highlight this limitation, **we have added a discussion in the future directions section on hallucination diagnosis and root-cause attribution (Section 7.1),** highlighting the need for frameworks that identify the earliest failure stage in the Vid-LLM pipeline responsible for the final unsupported output.
> >
> > Importantly, we believe **the taxonomy remains operationally useful** even without a complete disentanglement framework. In practice, overlapping failures can usually be narrowed down to a small set of candidate hallucination types rather than all categories simultaneously. For example, a temporal inconsistency caused by long-context memory degradation can be identified as involving temporal and long-context hallucination mechanisms, even if their exact causal relationship is unclear. This level of diagnosis is already sufficient to guide mitigation, since the relevant intervention families are correspondingly limited. Therefore, while future root-cause attribution frameworks could further improve classification consistency and interpretability, the proposed taxonomy already provides a practical mechanism for organizing failures and selecting candidate mitigation strategies.

---

> ### Author Response · Authors · 2026-06-26
>
> ## [RC 4] Provide higher-level insights and trade-offs in the mitigation section.
> We have substantially reorganized the mitigation material to emphasize synthesis over enumeration.
>
> ### 1. New synthesis section §5.6 "Effectiveness on Different Hallucination Types."
> Before the per-category method review, **§5** now distills the literature into five foundational method families—RL & preference optimization, supervision augmentation, structured representations, long-context information management, and inference-time intervention—organized according to where they intervene in the pipeline. **§5.6** then cross-cuts these families against the six hallucination types in **Table 3,** using a 5×6 matrix to rate how strongly each family mitigates each hallucination type, with ✓✓✓ indicating strong mitigation, ✓✓ indicating moderate mitigation, ✓ indicating limited mitigation, and ✗ indicating negligible mitigation. Each rating is derived from a qualitative assessment of the surveyed methods in the main text, based on two factors: research prevalence, namely how widely that family is used to address the hallucination type, and reported effectiveness, namely the gains reported by methods in that family. The accompanying discussion draws out several high-level patterns: structured representations are the most pervasive remedy; RL is broadly applicable and dominant for temporal and causal-reasoning hallucinations; long-context management is specialized and rarely repurposed; synthetic supervision contributes mainly to temporal hallucination; inference-time intervention contributes most to causal/reasoning hallucination; and a clear gap exists in the cross-modal column, which is addressed almost exclusively by representation-level alignment.
>
> ### 2. Per-paragraph trade-off statements.
>
> In **§6,** we add to each method paragraph some explicit statements of how strongly that family contributes to the target hallucination type and why, connecting the discussion back to the **Table 3** ratings (e.g., why verifiable rewards such as tIoU make RL one of the strongest levers for temporal hallucination, but only a secondary lever for object hallucination).
>
> Taken together, these additions shift the section from a sequential list of papers toward a structured account of which method family is most useful under which condition. This provides a more practical way for researchers to pick the corresponding mitigation technique for a given hallucination type, and helps them understand which technique is better and why some techniques are more effective than others.

---

> ### Author Response · Authors · 2026-06-26
>
> ## [RC 5] Use consistent benchmark while introducing mitigation techniques.
> We agree that comparing numbers drawn from different benchmarks, baselines, and metrics is misleading, and we address this in two coordinated steps.
>
> First, we add a new section, **§4 "Benchmarks and Evaluation"**, that shows, for each of the six hallucination types, the benchmarks used to evaluate it (organized per type in **§4.1–§4.6** and summarized in **Table 2** ). This makes the evaluation landscape explicit and clarifies, for each hallucination type, which datasets and evaluation formats are standard.
>
> Second, building on this section, we update the mitigation techniques in **§6** so that, when introducing methods with same kind, their results are reported on the **same benchmark** within each comparison rather than on heterogeneous ones. For example, Sparrow and TimeWarp are now both reported on TempCompass, so their reported gains can be read against the same reference. A few non-comparable "hallucination-rate reduction" figures that could not be tied to a common protocol were removed. Anchoring each method’s reported results to the benchmarks established in **§4** makes cross-method comparison within each hallucination type consistent and clear.
>
> ## [RC 6] Remove informal phrasing and unclear sentences.
>
> We have carried out a writing pass across the manuscript to remove informal phrasing, fix grammar issues, and reduce repetition—for example, replacing colloquial verbs with precise wording, tightening unclear sentences, and removing redundant restatements. We will continue to polish the prose in the camera-ready version and are happy to address any specific passages the reviewer flags.